

# Comparison of equatorial wave activity in the tropical tropopause layer and stratosphere represented in reanalyses

Young-Ha Kim[1], George N. Kiladis[2], John R. Albers[2,3], Juliana Dias[2,3], Masatomo Fujiwara[4], James A. Anstey[5], In-Sun Song[6], Corwin J. Wright[7], Yoshio Kawatani[8], François Lott[9], and Changhyun Yoo[10]

[1]Severe Storm Research Center, Ewha Womans University, Seoul, South Korea
[2]Physical Sciences Division, NOAA/Earth System Research Laboratory, Boulder, Colorado, USA
[3]Cooperative Institute for Research in Environmental Sciences, University of Colorado Boulder, USA
[4]Faculty of Environmental Earth Science, Hokkaido University, Sapporo, Japan
[5]Canadian Centre for Climate Modelling and Analysis, Environment and Climate Change Canada, Victoria, British Columbia, Canada
[6]Korea Polar Research Institute, Incheon, South Korea
[7]Centre for Space, Atmospheric and Oceanic Science, University of Bath, Bath, UK
[8]Japan Agency for Marine-Earth Science and Technology, Yokohama, Japan
[9]Laboratoire de Météorologie Dynamique, Ecole Normale Supérieure, Paris, France
[10]Department of Climate and Energy Systems Engineering, Ewha Womans University, Seoul, South Korea

**Correspondence:** Y.-H. Kim (kim@iau.uni-frankfurt.de), C. Yoo (cyoo@ewha.ac.kr)

**Abstract.** Equatorial Kelvin and mixed Rossby-gravity (MRG) waves in the tropical tropopause layer and stratosphere represented in recent reanalyses for the period of 1981–2010 are compared in terms of spectral characteristics, spatial structures, long-term variations and their forcing of the quasi-biennial oscillation. For both wave types, the spectral distributions are broadly similar among most of the reanalyses, while the peak amplitudes exhibit considerable spread. The longitudinal distri-
butions and spatial patterns of wave perturbations show reasonable agreement between the reanalyses. A few exceptions to the similarity of the spectral shapes and spatial structures among them are also noted. While the interannual variations of wave activity appear to be coherent for both the Kelvin and MRG waves, there is substantial variability in long-term trends among the reanalyses. Most of the reanalyses which assimilate satellite data exhibit large increasing trends in wave variance (∼15–50% increase in the 30 years at 100–10 hPa), whereas one reanalysis (JRA-55C) produced without satellite data does not. Several
discontinuities are found around 1998 in the time series of the Kelvin and MRG wave variances, which manifest in different ways depending on the reanalysis, and are indicative of impacts of the transition of satellite measurements during that year. The equatorial wave forcing of the quasi-biennial oscillation, estimated by the Eliassen–Palm flux divergence, occurs in similar phase-speed ranges among the reanalyses, while the forcing magnitudes show considerable spread. The forcing maxima of the Kelvin waves exhibit slightly different altitudes between the reanalyses (by ∼3 km at around 15 hPa). In addition, at around
20 hPa, a wave signal which appears only in easterly mean winds with westward phase speeds is found and discussed.



## 1 Introduction

Stratospheric equatorial waves are known to be generated in response to the heat sources associated with tropical convection and to play an important role in the tropics (Salby and Garcia, 1987; Garcia and Salby, 1987). On subseasonal time scales, Kelvin waves lead to large variations in tropopause temperature and height (e.g., Tsuda et al., 1994; Kim and Son, 2012; Kim and Alexander, 2015) and modulate the formation of cirrus clouds (Boehm and Verlinde, 2000; Immler et al., 2008). Kelvin waves in the tropical tropopause layer (TTL) are also important in stratosphere–troposphere exchange, as they modulate the amount of water vapor entering the stratosphere via dehydration of air and transport and mixing of chemical species such as ozone via wave breaking (Fujiwara et al., 1998, 2001; Plumb, 2002; Fueglistaler et al., 2009). It has been observed that mixed Rossby-gravity (MRG) wave circulations in the lower troposphere are related to tropical cyclogenesis (Dickinson and Molinari, 2002; Zhou and Wang, 2007). In the stratosphere, the Kelvin and MRG waves, along with smaller-scale waves, are known as sources of momentum needed to drive the easterly-to-westerly and westerly-to-easterly phase transitions of the quasi-biennial oscillation (QBO), respectively (Holton and Lindzen, 1972; Dunkerton, 1997; Baldwin et al., 2001), although the momentum transported by the MRG waves needs to be further quantified (Randel et al., 1990; Kim and Chun, 2015).

Investigation of the distribution and variability of large-scale equatorial waves requires datasets with global coverage and global (re)analyses are extremely useful datasets for this purpose. A reanalysis is a product of a data assimilation system which reconciles the observed atmospheric states from many kinds of measurements with the atmospheric governing equations resolved by a numerical prediction model. Reanalysis products depend on the assimilation method and prediction ("first guess") model, especially in the stratosphere where the assimilated fields are less constrained by the observations due to less density of observations when compared to the troposphere. Around the TTL, there exist abrupt vertical changes in the temperature and stability, which greatly modify characteristics of the equatorial waves, particularly their vertical wavelengths and amplitudes (e.g, Randel and Wu, 2005; Ryu et al., 2011). Representation of the TTL in reanalyses might be sensitive to the vertical resolution of the prediction model and assimilation techniques used (see Birner et al., 2006, for a case of the extratropical tropopause). Therefore, it is important to identify the difference/spread among various reanalysis products in their representation of the equatorial waves in the TTL and stratosphere. For example, equatorial wave activity in the TTL during 1990–1999 was compared using seven reanalyses by Fujiwara et al. (2012) and was found to vary significantly among them.

In contrast to analyses, which are made using operationally changing, state-of-the-art versions of the prediction model and assimilation system, reanalyses are derived using fixed versions of the data assimilation system and first guess models for the whole period of the product. This helps to generate a temporally homogeneous product, which benefits studies of long-term changes in meteorological variables. However, reanalyses also have a potential for inhomogeneities or discontinuities arising from introduction of new observational data into the assimilation. Examples include the introduction of radiance/temperature profiles derived from the Television Infrared Observation Satellite (TIROS) Operational Vertical Sounder (TOVS) suite (Smith et al., 1979) around 1978 and those from the Advanced TOVS (ATOVS) suite around 1998. It has been reported that the transition from the TOVS to ATOVS suites induces temporal discontinuity in assimilated variables such as up-



per stratospheric global-mean temperature (Onogi et al., 2007; Simmons et al., 2014) and equatorial stratospheric mean wind and temperature (Kawatani et al., 2016), although the transition generally improved the quality of reanalyses after about 1998.

To the best of our knowledge, impacts of such satellite transitions in reanalyses have so far not been studied in the context of equatorial waves. It is likely more difficult to identify discontinuities or inhomogeneities in wave fields which, by definition,

contain smaller-scale variations in space and time than mean fields. Recently, a pair of reanalyses have been identically produced using a single assimilation system, with the exception that the satellite data are assimilated in one (JRA-55) but not in the other (JRA-55C; see Table 1 for the abbreviations and references). These datasets can allow us to identify the impact of satellite data in that assimilation system and, in particular, to help further distinguish the impact of the TOVS–ATOVS transition.

In this study, we investigate the characteristics of equatorial Kelvin and MRG waves in the TTL and stratosphere and how

they differ between recent reanalyses for the period 1981–2010 (Section 3.1). Spatial distributions and patterns of the waves in the reanalyses are presented in Section 3.2. In Section 3.3, long-term changes in equatorial wave amplitudes are compared among the reanalyses and, based on the comparison between JRA-55 and JRA-55C, the effects of the satellite data upon the assimilated waves are discussed. In addition, spectra of the Eliassen–Palm (EP) flux and its divergence, a measure of wave–mean flow interaction, are presented to compare the equatorial wave forcing of the QBO estimated from different reanalyses

(Section 3.4). Among the results presented in Section 3.1, we identify a wave spectrum that has not been dealt with in the literature, found at $20\,\mathrm{hPa}$ along with the well-documented equatorial waves in all the reanalyses studied. This wave spectrum is further discussed in Section 4. A summary of the results is included in Section 5.

## 2   Data and method

We examine six reanalyses: ERA-I, MERRA, MERRA-2, CFSR, JRA-55, and JRA-55C (see Table 1 for their full names and

key citations). The horizontal and vertical winds, temperature, and geopotential in the TTL and stratosphere for the period of 1981–2010 are used. The data we use are stored at 3-hour intervals for MERRA and MERRA-2 and 6-hour intervals for the others. The results do not significantly change if we use 6-hourly subsampled time series for MERRA and MERRA-2 (not shown) as we analyze equatorial waves with periods longer than 2 days. Both pressure-level datasets, which are interpolated to standard levels (SL, e.g., $100\,\mathrm{hPa}$), and model-level (ML) datasets are used for each reanalysis, except for MERRA of which

ML data are not available. At $100\,\mathrm{hPa}$ and above in the tropics, pressure variations on a model level are negligibly small or absent in all of the reanalyses used in the present study (see Supplement to Fujiwara et al., 2017). Therefore, the model levels can be regarded as being at nearly constant pressure levels, which allows us to simply perform spectral calculations on horizontal planes without the need to introduce vertical interpolation. For spectral shapes and spatial distributions of Kelvin and MRG waves, the results were very similar between SL and ML datasets (not shown), whereas wave amplitudes differ

significantly as will be seen in Fig. 8. Therefore, only the SL results are presented in Sections 3.1 and 3.2, and the ML results (or both) are presented for quantitative analysis of wave variances and EP flux profiles in Sections 3.3 and 3.4.

Zonal wavenumber–frequency ($k$–$\omega$) spectra are calculated monthly at each latitude and height for each of the symmetric and antisymmetric components of variables with respect to the equator. To obtain the monthly spectra, we use a 90-day time





window centered on the target month. The window function is defined as $C$ for the central 30 days and $C\sin(\pi t/60)$ and $C\cos(\pi t/60)$ for the first and last 30 days, respectively, where $t$ is the time (in days) relative to the first day of each 30-day segment and the normalization constant $C = \sqrt{3/2}$. The window function is determined such that for a long-term mean, the integral of the power spectrum equals the unwindowed variance of the variable. The spectra are calculated using the Fourier

transform after removal of zonal mean and application of the time window, and they are averaged over the latitude range 15°N–15°S. The spectra are plotted in a variance-preserving form using base-10 logarithm axes in frequency and wavenumber.

The use of a 90-day time window retains intraseasonal variations such as the MJO partly in the spectra in the TTL at the lower-frequency range, as will be seen in Fig. 1. However, we will exclude these signals from our analysis and focus on signals with periods shorter than 20 days.

Following previous studies (Wheeler and Kiladis, 1999; Hendon and Wheeler, 2008; Fujiwara et al., 2012), the background spectra are obtained using the 1-2-1 filter repeatedly in wavenumber and frequency for each reanalysis. In this study, common background spectra for symmetric and antisymmetric components are obtained by averaging spectra of the two components before applying the filter. The filter is applied to the logarithm of the power spectrum; the number of passes is 23 for zonal wavenumber and 7 for frequency (Fujiwara et al., 2012).

$k$–$\omega$ spectra of the EP flux are also calculated monthly for symmetric and antisymmetric modes (Section 3.4). The EP flux formulation defined with the transformed Eulerian mean of the primitive equations (Eq. 3.5.3 in Andrews et al., 1987) is used. The EP flux spectra for symmetric (antisymmetric) modes are obtained using $u_S$, $T_S$, $w_S$, and $v_A$ ($u_A$, $T_A$, $w_A$, and $v_S$), where $u$, $v$, $w$, and $T$ are the zonal, meridional, and vertical winds and temperature, respectively, and the subscripts S and A denote the symmetric and antisymmetric components of each variable, respectively. The same window function as the above is applied

to all variables before calculating cospectra. In the figures, the EP flux is divided by the mean radius of the Earth, so that it has units of Pascals.

The buoyancy frequency used to compute equivalent depths of the equatorial waves is set to $0.024\,\mathrm{s}^{-1}$ based on the climatological temperature profile of the tropical lower stratosphere (e.g., Grise et al., 2010).

## 3   Results

### 3.1   Spectral characteristics

Figure 1 shows $k$–$\omega$ power spectra for the eastward-propagating, symmetric component of $T$ ($T_S$) at 100, 70, 50, and $20\,\mathrm{hPa}$ from the SL datasets of the six reanalyses, averaged over the 30-year period (1981–2010). The major portion of the spectral power appears along the Kelvin wave dispersion curves (black solid lines) at all altitudes. The spectral characteristics are broadly similar among the reanalyses with peaks at $k = 2$–3. The reanalyses commonly exhibit a gradual shift of the Kelvin

wave spectrum to larger equivalent depths ($h$) with respect to the altitude: the majority of the spectral power appears at $h < 60\,\mathrm{m}$ at $100\,\mathrm{hPa}$, at $h \sim 60\,\mathrm{m}$ at $70\,\mathrm{hPa}$, and at $h = 60$–$240\,\mathrm{m}$ above $70\,\mathrm{hPa}$. There exist variations in the spectral width in $h$: for example, compared to the other reanalyses, CFSR shows broader spectra at small equivalent depths ($h < 60\,\mathrm{m}$) consistently at $70$–$20\,\mathrm{hPa}$, with peaks at slightly smaller equivalent depths ($70$–$50\,\mathrm{hPa}$). In addition to the Kelvin wave signal, at $100\,\mathrm{hPa}$,





another peak exists at $k = 2$ in the low-frequency range ($\omega < 0.04\,\mathrm{cyc\,day}^{-1}$). As mentioned in Section 2, this is likely related to the intraseasonal MJO (Hendon and Wheeler, 2008).

The relative magnitude of Kelvin wave spectral power between ERA-I, MERRA, MERRA-2, and CFSR varies with height (Fig. 1), while JRA-55 and JRA-55C show generally less power below $20\,\mathrm{hPa}$ compared to the other four datasets. Note that
relatively small variances in JRA-55 and JRA-55C are found in the temperature field but not in the wind fields (see Fig. S1). In Fig. 1, the ratio of the spectral power to the background spectrum is also shown (thin purple), which indicates statistical significance of the spectral signals. A ratio of 1.2 was deemed to be statistically significant at the 95% level by Wheeler and Kiladis (1999), and for the majority portions of the Kelvin wave spectra in Fig. 1, the ratios are generally larger than 1.5 (i.e., 50% larger than the background spectral power), conservatively implying that the spectral peaks are all statistically significant.

Figure 2 shows $k$–$\omega$ spectra for the westward-propagating, symmetric component of $v$ ($v_\mathrm{S}$) at 100, 70, 50, and $20\,\mathrm{hPa}$. Note that the lower bound of $y$-axis at $100\,\mathrm{hPa}$ is different from that at the other levels, as the $v_\mathrm{S}$ spectrum is much broader in frequency at $100\,\mathrm{hPa}$. The 100-hPa spectrum has periods from around 2.5 days ($0.4\,\mathrm{cyc\,day}^{-1}$) to longer than 30 days ($0.033\,\mathrm{cyc\,day}^{-1}$), with a peak at $k = -5$, $\omega = 0.1$–$0.2\,\mathrm{cyc\,day}^{-1}$ common to all of the reanalyses. In the upper troposphere, the background zonal wind near the equator is westerly (easterly) in the western (eastern) hemisphere. The MRG waves
generated in the region of westerly (easterly) background flow can have relatively low (high) ground-based frequencies, which might result in the broad spectrum in frequency of $v_\mathrm{S}$ at $100\,\mathrm{hPa}$. The low-frequency portion of the wave spectrum at $100\,\mathrm{hPa}$ seems to roughly follow the MRG wave dispersion curves for a background wind ($U$) of $+10\,\mathrm{m\,s}^{-1}$ (Fig. 2, dashed).

At $70\,\mathrm{hPa}$, the low-frequency portion ($< 0.1\,\mathrm{cyc\,day}^{-1}$) of the spectrum is mostly filtered out, and commonly in all reanalyses, the peaks appear at $k = -6$, $\omega \sim 0.20\,\mathrm{cyc\,day}^{-1}$ and at $k = -5$, $\omega \sim 0.22$–$0.25\,\mathrm{cyc\,day}^{-1}$ ($h \sim 60\,\mathrm{m}$ for both, assuming
$U = 0$) (Fig. 2). However, the lower bound in frequency of the spectral power is different between the reanalyses: JRA-55C, JRA-55, and MERRA-2 exhibit broader frequency spectra than the others. Above $70\,\mathrm{hPa}$, two additional peaks appear at $k = -4$, $\omega \sim 0.3\,\mathrm{cyc\,day}^{-1}$ and at $k = -3$, $\omega \sim 0.4\,\mathrm{cyc\,day}^{-1}$, which are more intense than those at lower frequencies with $|k| > 4$, as the altitude increases.

Figure 2 also exhibits a distinct feature in the $v_\mathrm{S}$ spectra at $20\,\mathrm{hPa}$, compared to the spectra at the lower altitudes: statistically
significant power appears along the narrow spectral region that includes $k = -5$, $\omega \sim 0.5\,\mathrm{cyc\,day}^{-1}$ and extends into higher wavenumbers and frequencies in all of the reanalyses. Toward lower wavenumbers, it seems to merge into the aforementioned peak at $k = -3$, $\omega \sim 0.4\,\mathrm{cyc\,day}^{-1}$. The spectral power along this region is larger in JRA-55C, JRA-55, and CFSR than in the others. This portion of the spectrum is further examined in Section 4. As will be seen therein, the waves with this spectrum do not originate from below, and they appear with different timing and different characteristics from the lower-frequency
upward-propagating MRG waves which dominate the $v_\mathrm{S}$ spectra in the lower altitudes. In Sections 3.2–3.4, we focus on the lower-frequency MRG waves filtered with a cut-off frequency of $0.33\,\mathrm{cyc\,day}^{-1}$ (period of 3 days).

The spectral shapes of Kelvin and MRG waves obtained from the ML datasets (not shown) are very similar to those from the SL datasets (Figs. 1 and 2) for each reanalysis, whereas their spectral power is larger by up to about 35%, depending on the altitude and reanalysis. This will be further discussed in Section 3.3.





In the following sections, we define Kelvin waves as the symmetric mode with $h = 8$–$240\,\mathrm{m}$, $k = 1$–$10$, and periods of 2–20 days following Fujiwara et al. (2012), unless otherwise stated. These spectral components include a major portion of the Kelvin wave variances (Fig. 1) while excluding contributions of the other disturbances at low phase-speed ranges at $100\,\mathrm{hPa}$ (see also Fig. 9). The MRG waves are defined as the antisymmetric mode with $h > 8\,\mathrm{m}$, $-10 \leq k < 0$, and periods of longer than 3 days, as previously mentioned, where $h$ is for $U = 0$. The perturbations filtered for these spectral components are denoted as, for example, $T_{\mathrm{Kelvin}}$ for the Kelvin wave temperature or $v_{\mathrm{MRG}}$ for the MRG wave meridional wind.

### 3.2 Spatial structure

To investigate the spatial distributions, the Kelvin and MRG signals are reconstructed in physical space by filtering their spectral components. Figure 3a shows the distributions of $T_{\mathrm{Kelvin}}$ and $v_{\mathrm{MRG}}$ variances at 100, 70, 50, and $20\,\mathrm{hPa}$ during the period of 1981–2010, averaged for ERA-I, MERRA-2, CFSR, and JRA-55 (the reanalyses that assimilate satellite data are included as the ensemble members, but MERRA is excluded since it exhibits very similar spatial distributions to MERRA-2, as will be seen in Fig. 3b). The ensemble-mean variances are normalized by their maximum values on each horizontal plane. Both the Kelvin and MRG wave variances are confined near the equator, and the locations of their maxima slant eastward in the vertical, consistent with the equatorial wave theory. Note that, while the phase of MRG waves propagates westward, their wave packet travels eastward. The maxima of variances for the Kelvin (MRG) waves are located in the eastern (western) hemisphere in the lower stratosphere: 70 and $50\,\mathrm{hPa}$, consistent with previous observational studies (Alexander et al., 2008; Yang et al., 2012; Kiladis et al., 2016). At $20\,\mathrm{hPa}$, the Kelvin and MRG waves show rather broad distributions in longitude. The distributions of the wave variances at $100\,\mathrm{hPa}$ are closely related to those of background zonal wind in the upper troposphere (Yang et al., 2012; Flannaghan and Fueglistaler, 2013). The easterly (westerly) upper tropospheric wind in the eastern (western) hemisphere allows Kelvin (MRG) waves to more readily propagate vertically, resulting in the hemispheric difference in the wave variances. In addition, the MRG wave variances at $100\,\mathrm{hPa}$ are distributed quite broadly in longitude (Fig. 3a). Yang et al. (2012) showed that during the northern winter the MRG wave variances in the upper troposphere are much larger in the western hemisphere with peaks at about $140°\mathrm{W}$ and $30°\mathrm{W}$ than in the eastern hemisphere, whereas during the northern summer, the variance distribution becomes broad, stretching to the western Pacific (see their Fig. 6).

Figure 3b shows the wave variances at the equator in each of the six reanalyses. The longitudinal variations of the Kelvin wave variances seem broadly similar among the reanalyses, with peaks at approximately $45°\mathrm{E}$ at $100\,\mathrm{hPa}$ and at $75$–$110°\mathrm{E}$ at $50\,\mathrm{hPa}$, although the Kelvin waves in CFSR show much smaller longitudinal variations than those in the other reanalyses at 100 and $50\,\mathrm{hPa}$. The MRG waves have maximum variances in the eastern Pacific and South America at 70 and $50\,\mathrm{hPa}$, and these maxima are roughly twice the minima over the Maritime Continent and western Pacific, common to all of the reanalyses. However, at $50\,\mathrm{hPa}$, detailed distributions in the eastern hemisphere show some differences among the reanalyses: the peak in ERA-I is eastward of that in the others (e.g., $35$–$45°$ eastward of CFSR, JRA-55, and JRA-55C), and the variance in CFSR seems to extend further to the west ($\sim150°\mathrm{W}$). Differences between JRA-55 and JRA-55C appear over the Indian ocean for Kelvin waves and the eastern Pacific for both waves, whereas the differences are very small on the Maritime Continent and western Pacific at all altitudes. These could be explained in part by relatively large numbers of radiosonde observations near





the Maritime Continent compared to the other regions around the equator in these two reanalyses (Fig. 5 in Kawatani et al., 2016; Figs. 2-5c and 2-6c in Wright et al., in preparation).

The currently used filters for the MRG waves exclude the low-frequency perturbations ($< 0.1\,\mathrm{cyc\,day}^{-1}$) at 100 hPa revealed in Fig. 2, of which the spectrum follows the dispersion curves for $U \sim 10\,\mathrm{m\,s}^{-1}$. An additional calculation, the same as with

Fig. 3, is performed but for the 100-hPa low-frequency components of $v_\mathrm{S}$ using the filters for $0.033 \leq \omega < 0.1\,\mathrm{cyc\,day}^{-1}$ and $-10 \leq k < 0$ (Fig. S2). It is observed in all the reanalyses that the low-frequency perturbations are located mostly in the western hemisphere where westerlies exist (over the eastern Pacific and Atlantic), consistent with the Doppler shifted dispersion curves discussed in Fig. 2.

To further investigate the spatial structures of the Kelvin and MRG waves, including the circulation patterns and hori-

zontal scales of representative wave modes in each reanalysis, an EOF analysis is used following the technique outlined in Kiladis et al. (2016). In that study, EOFs were calculated from the covariance matrix of a 2–6 day filtered meridional wind in 20°N–20°S from ERA-I data to isolate MRG waves. The filter band was based on the well-documented strong spectral peak in the equatorial meridional wind centered at around the 4.5-day period (Fig. 2). Dynamical fields associated with each EOF were obtained by projecting unfiltered ERA-I data at each grid point onto the associated principal component (PC) time series.

Here we use data interpolated to 2.5° resolution for the six reanalyses considered. A similar technique as in Kiladis et al. (2016) is used to obtain the statistical structures of the equatorial Kelvin waves, except that a 2–25 day eastward-only filter band is applied to the equatorial zonal wind based on the spectral peaks in Fig. 1. The results are very robust to changes in the filtering, as long as the filter band contains the spectral peaks shown in Figs. 1 and 2, and to changes in latitudinal extent of the EOF basis.

In all cases, EOF "pairs" are obtained with respective PC time series that correlate at better than 0.96, which together represent the propagating pattern of Kelvin or MRG waves. Thus, each mode can be represented by either EOF pattern and its associated PC. Figure 4 shows the projected structures of the leading modes (EOF-1) of 50-hPa Kelvin waves from the six reanalyses for the period of 1981–2010. Wind vectors are shown at the locations where they are statistically significant, taking into account temporal and spatial autocorrelation. In all cases, zonal wavenumber 1 structures are obtained as the leading

modes in the tropics, with zonal wind perturbations in phase with geopotential, as expected from theory. Especially in the tropics, remarkably similar patterns are found in all of the reanalyses. The wind and geopotential perturbations exhibit much larger amplitudes in the Eastern Hemisphere than in the Western Hemisphere, consistent with the result in Fig. 3. Higher-order Kelvin EOF pairs represent integer zonal wavenumber structures, with $k = 2$ Kelvin structures comprising the second EOF pair (EOFs 3 and 4). The second EOF pairs also show reasonable agreement among the reanalyses (not shown). The $k = 1$

patterns shown in Fig. 4 account for the largest portion of total variance in the equatorial 2–25 day eastward zonal wind in each reanalysis, which amounts to a maximum of 42% in the case of ERA-I and somewhat smaller amounts in the other reanalyses (Table 2).

Lag regressions based on the PC time series show the eastward propagation of the Kelvin wave signals at a mean phase speed of around $35\,\mathrm{m\,s}^{-1}$ (not shown), which is rather faster than the $20$–$30\,\mathrm{m\,s}^{-1}$ found in previous studies, although these

phase speeds are highly dependent on the state of the QBO (e.g., Randel and Wu, 2005; Ern et al., 2008; Lott et al., 2009,





2014). While these features will be explored in further detail in a future paper, the main point here is that the various reanalyses appear to be quite suitable for studying the mean statistical structure of these waves, as well as their variability, as will be demonstrated below.

The situation for the MRG wave signals is rather different (Fig. 5). The leading MRG EOFs represent localized wave packets
that are confined over the eastern Pacific to Atlantic sector, as expected from the location of the variance peaks in Fig. 3. The EOF patterns have classical structures of equatorial gyres and off-equatorial antisymmetric geopotential perturbations as predicted by Matsuno (1966). The reanalyses each represent structurally very similar MRG wave patterns, except for CFSR which has a zonally broader signal (i.e., longer wavelengths). In addition, the MRG wave patterns in JRA-55, JRA-55C, and CFSR are displaced somewhat to the west of the other reanalyses, again reflecting their variance distributions shown in Fig. 3b.
There is less agreement among the reanalyses in the higher mode MRG EOFs (not shown).

Kelvin and MRG signals at other levels from 70 to $10\,\text{hPa}$ are broadly similar to those shown here, although the structures change considerably at $100\,\text{hPa}$ (not shown). In summary, there is reasonable agreement between the reanalyses in their statistical representation of Kelvin and MRG wave structures, at least for the leading modes.

### 3.3 Long-term change and satellite effects

In this section, we analyze long-term variations in the Kelvin and MRG wave activity and discuss the impact of satellite data upon the assimilated wave activity based on the comparison of JRA-55 and JRA-55C. Figure 6 shows annual-mean time series of 100-hPa $T_{\text{Kelvin}}$ and $v_{\text{MRG}}$ variances, as defined in Section 3.1. The results from the ML datasets are presented (solid), except for MERRA for which the SL datasets are used (dashed) due to the data availability. The variance of $T_{\text{Kelvin}}$ in ERA-I fluctuates between about 0.5 and $0.85\,\text{K}^2$ with an increasing trend, whereas that in JRA-55C fluctuates between ~0.35
and $0.58\,\text{K}^2$ without such an obvious trend. Despite this difference, much of the interannual variability in the Kelvin wave activity is reflected in both time series, with a Pearson linear correlation between the two of 0.85 (0.88 after trend removal). Similar correlations are obtained between the other series, with the variances ranging between those in ERA-I and JRA-55C. By comparing JRA-55 and JRA-55C, it appears that assimilation of satellite data leads to an increase in the Kelvin wave temperature variance, as also found in the previous section (Fig. 3b).

The difference in the variance between JRA-55 and JRA-55C is further investigated in Fig. 7, which presents the annual time series of the difference at altitudes from 100 to $5\,\text{hPa}$. Because the wave variance in the stratosphere is strongly dependent on the QBO (e.g., Yang et al., 2012), 1-2-1 smoothing is applied to the annual time series to filter out the quasi-biennial fluctuations and focus on the longer-term variations. It is evident that the difference (JRA-55 minus JRA-55C) in the $T_{\text{Kelvin}}$ variance is generally positive from 100 to $20\,\text{hPa}$ during the 30-year period, indicating the enhancement of the Kelvin wave amplitude
by assimilation of satellite data in JRA-55. In addition, it is noteworthy that at $100\,\text{hPa}$, the difference is systematically larger from around 2000, compared to that before late 1990s: it is roughly 6% (up to ~8%) in the years before 1998, whereas it increases to ~10% in 1998–2000 and becomes ~20% afterward. A similar systematic change of the difference in the $T_{\text{Kelvin}}$ variance is even more evident in the upper stratosphere: at 10 and $5\,\text{hPa}$ (Fig. 7) and above (not shown), where the difference is mostly negative until 1998 and becomes positive after 1999.





The systematic change in the impact of satellite data assimilation on the Kelvin wave amplitude in JRA-55 around 1998 might be due to the TOVS–ATOVS transition, given the timing of the change (see Fig. 8 in Fujiwara et al., 2017, for the timelines of satellite data used in JRA-55 and other reanalyses). The Advanced Microwave Sounding Unit A (AMSU-A) instruments in the ATOVS suite were introduced in 1998. Compared to the Stratospheric Sounding Unit (SSU) instruments

in the TOVS suite, the AMSU-A observations have better vertical coverage with a higher vertical resolution (see Fig. 7 in Fujiwara et al., 2017, for the vertical weighting functions of the SSU and AMSU-A measurements). The vertical weighting functions of the two suggest that the AMSU-A instruments could be advantageous over the SSU, particularly at an altitude of $\sim$100 hPa which the latter does not cover. This may explain the systematic change found at 100 hPa around 1998 (Fig. 7). The higher vertical resolution of the AMSU-A instruments is expected to mostly benefit the upper stratosphere where radiosonde

sounding observations do not reach (i.e., above $\sim$10 hPa). This is consistent with the observed systematic change being most prominent in the upper stratosphere. In addition to the TOVS–ATOVS transition, there is also a possibility that the Global Navigation Satellite System Radio Occultation (GNSS-RO) has influenced the assimilated waves in JRA-55 since 2006 (Fig. 10 in Fujiwara et al., 2017), although this cannot be identified by the current analysis.

Recalling that JRA-55C does not exhibit a long-term trend in Kelvin wave activity at 100 hPa (Fig. 6), the trend shown in

JRA-55 is probably not an actual change in the true atmosphere, but instead an artifact arising due to the satellite transition. The rate of change in the 100-hPa $T_{\mathrm{Kelvin}}$ variance around 1998 is 17% in JRA-55 when it is measured by the difference between the variances averaged for the two periods before and after 1998, i.e., 1981–1997 (P1 hereafter) and 1999–2010 (P2), relative to the 30-year mean variance. The rates of changes in the other reanalyses are comparable to or smaller than that in JRA-55 (e.g., 19% in ERA-I, which is the largest value). This suggests that the long-term changes in the 100-hPa Kelvin wave activity

shown in those reanalyses also could result largely from the satellite transition.

The variance of $v_{\mathrm{MRG}}$ exhibits generally similar interannual variations at 100 hPa between all of the reanalyses (Fig. 6), while the magnitudes of the variance are different. The $v_{\mathrm{MRG}}$ variance has an increasing long-term trend from the early 1990s, even in JRA-55C. The $v_{\mathrm{MRG}}$ variance at 100 hPa in JRA-55 is always larger than that in JRA-55C by roughly 6–12% (Figs. 6 and 7), reflecting the impact of satellite data on the assimilated MRG waves. In addition, the assimilation of satellite data

increases the analyzed MRG wave activity at 5 hPa in JRA-55 (Fig. 7). This increase is up to $\sim$55% and it becomes even larger at higher altitudes (see Fig. 8). Notable differences between P1 and P2 in the satellite impact on the assimilated MRG wave activity are not found below 5 hPa (Fig. 7). At 5 hPa, the impact is significantly larger in P2 than in P1 (note that two different scales are used in the $y$-axis in the bottom panel of Fig. 7, below and above 20%, separated by the dashed horizontal line).

Figure 8 presents the vertical profiles of $T_{\mathrm{Kelvin}}$ and $v_{\mathrm{MRG}}$ variances averaged for P1 and P2 along with the differences

between the two periods. In the ML results, the $T_{\mathrm{Kelvin}}$ variance is maximized at approximately 80 hPa in ERA-I and at 70 hPa in the others for both periods. The $T_{\mathrm{Kelvin}}$ variance decreases with height from these levels to $\sim$30 hPa, and gradually increases from $\sim$20 hPa in both periods in all the reanalyses, except in MERRA-2 during P1. Further examination indicates that sometimes the temperature seems not to be properly assimilated in the upper stratosphere during P1 in MERRA-2 (Fig. S3; note the lack of Kelvin wave peaks in 1984, 1989, and 1991), which may result in the decrease above $\sim$20 hPa in P1 in MERRA-2.

It has previously been reported that the monthly mean zonal wind at 10 hPa in MERRA-2 shows significant differences from





observations, in particular with larger annual/semiannual oscillations, until the mid-1990s (Coy et al., 2016; Kawatani et al., 2016), which is perhaps partly related to the under-representation of the Kelvin wave temperature shown in Fig. S3. Errors in a mean state of assimilated fields can cause degradation of any variables describing waves during the first-guess model integration, which in turn degrade the assimilated fields 3 or 6 hours later. In P2, the $T_{\mathrm{Kelvin}}$ variance at 70–30 hPa has similar

magnitudes between ERA-I, MERRA-2, and CFSR, while JRA-55 exhibits a relatively smaller variance than these reanalyses, as is also evident in Fig. 1. In the upper stratosphere, the variance is notably larger in ERA-I in P2 than in the others (also see Fig. S3).

The difference in the $T_{\mathrm{Kelvin}}$ variance between P1 and P2 is positive in the whole stratosphere in all of the reanalyses (Fig. 8). In JRA-55C, the difference is small at 100 hPa (∼6%), as already discussed (Fig. 6), and above 10 hPa (4–8%), although it

reaches ∼20% at 50–20 hPa. Given the small difference in Kelvin wave activity at ∼100 hPa, the increase in the activity at 50–20 hPa could result from interaction with the QBO which itself has variability in its morphology. For instance, if durations of westerly QBO phases below ∼50 hPa are shorter in P2 than in P1 on average (not shown), they could cause the increase in the Kelvin wave activity at 50 hPa and above in JRA-55C. However, since only five (seven) QBO cycles are included in P2 (P1), further investigation of the long-term change associated with the QBO might be less meaningful in a statistical sense.

Another possible reason could be the increased number of radiosonde observations around 30 hPa with time (see Fig. 15 in Kawatani et al., 2016). In JRA-55, the difference is large in the upper stratosphere (∼25% or larger), and it is even larger in ERA-I, MERRA-2, and CFSR. Similarly to the case of JRA-55, the large increase in Kelvin wave activity in the middle and upper stratosphere in the other reanalyses could also be in large part due to the transition between satellite instruments. From Fig. S3, it is evident that the difference in the Kelvin wave variances at 10 hPa between ERA-I and the others increases abruptly

around 2000, which may support such an impact of the satellite instrument change in ERA-I.

In all the reanalyses, the $v_{\mathrm{MRG}}$ variance decreases with height below ∼20 hPa and increases above ∼10 hPa (Fig. 8). In both periods, MERRA-2 and CFSR exhibit larger variances than the others, particularly in the upper stratosphere. In P1, it is seen in the upper stratosphere that the $v_{\mathrm{MRG}}$ variance in the ML result of CFSR varies less smoothly in the vertical and that the variance in MERRA-2 is about 2–3 times larger than that in the others. These features are still present to a lesser extent in P2.

Examination of time series of the $v_{\mathrm{MRG}}$ variance (Fig. S3) shows that some unexpected peaks appear in P1 in MERRA-2 with exceptionally large magnitudes even during the easterly QBO phase, which are primarily responsible for the large value of the $v_{\mathrm{MRG}}$ variance in P1 shown in Fig. 8. The unexpected peaks disappear suddenly starting in 1999 (Fig. S3), likely emphasizing the impact of the AMSU-A observations on the assimilated wave activity in MERRA-2. Regarding the vertical fluctuations exhibited in CFSR, the reason for this is not obvious. The profiles of the difference between P1 and P2 may imply that above

10 hPa, the large increase in the $v_{\mathrm{MRG}}$ variance in JRA-55 (10–25%) and ERA-I (∼40%), compared to the small increase in JRA-55C (∼5%), is probably in large part due to the satellite transition.

In each reanalysis, the variances obtained from the SL datasets are smaller than those from the ML datasets in both periods (Fig. 8), by 3–29% (0–38%) for Kelvin (MRG) waves depending on the altitudes and reanalyses. The differences in the variances between the ML and SL results (ML minus SL) normalized by the variances in the ML results at each altitude

are reported in Table 3. The smaller amplitudes of the waves in the SL datasets compared to those in the ML datasets result



from vertical interpolation of reanalysis output variables, which inevitably damps wave perturbations in the standard-level products of reanalyses. The damping by the interpolation is more significant for waves with smaller vertical wavelengths (e.g., Kim and Alexander, 2013; Kim and Chun, 2015). In addition, for a given pressure level, the damping of waves in the SL datasets also depends on the distance between the given level and its adjacent model level. That is, the interpolation effect must
be less if a model level is very close to the given pressure level. For example, the distance between $100\,\mathrm{hPa}$ and its adjacent model level in MERRA-2 is very small ($< 50\,\mathrm{m}$), and thus the difference between the SL and ML results of MERRA-2 is only 3–4% at $100\,\mathrm{hPa}$ (Table 3). The same (opposite) is true for the $10\,\mathrm{hPa}$ ($50\,\mathrm{hPa}$) in ERA-I, with a small (large) difference at that level. In JRA55 and JRA-55C, the difference in the $v_{\mathrm{MRG}}$ variance between the ML and SL results is less than 5% at all levels, implying that the ML fields of these two reanalyses contain less of the MRG wave perturbations with small vertical
wavelengths ($\sim 2\Delta z$–$4\Delta z$, where $\Delta z$ is the vertical grid spacings of their model), compared to the other reanalyses.

### 3.4   EP flux

In this section, we compare the vertical EP flux ($F_z$) spectra and the wave forcing of the QBO calculated by the EP flux divergence among the reanalyses. As for the figures discussed in Section 3.1, we obtain $k$–$\omega$ spectra for each of the symmetric and antisymmetric modes but for $F_z$ (Figs. 9 and 10, respectively). The spectral characteristics of $F_z$ for the symmetric and
antisymmetric modes are qualitatively similar to those of $T_{\mathrm{S}}$ and $v_{\mathrm{S}}$, respectively, for each reanalysis. For the symmetric mode, the $F_z$ spectra are aligned along the Kelvin wave dispersion curves with $h$ values that are similar to those for $T_{\mathrm{S}}$, along with exhibiting a gradual shift of the spectra to the higher $h$ with increasing altitudes (Figs. 1 and 9). On the other hand, the symmetric $F_z$ and $T_{\mathrm{S}}$ spectra differ in that the $F_z$ ($T_{\mathrm{S}}$) spectra have peaks at $k = 3$–$5$ ($2$–$3$) in all of the reanalysis and that the magnitudes of $F_z$ in JRA-55 and JRA-55C are not smaller compared to those in the others as was the case for temperature
(cf. Fig. 1). As already mentioned, JRA-55 and JRA-55C have relatively weak Kelvin wave temperature amplitudes but not wind amplitudes (Fig. S1), and the major term of $F_z$ for Kelvin waves is the vertical momentum flux ($w'u'$). The reason that the peak of $F_z$ occurs at larger zonal wavenumbers than in the $T_{\mathrm{S}}$ spectra is because the vertical wind perturbations tend to have large amplitudes at higher zonal wavenumbers: the ratio of vertical to horizontal wind amplitudes is proportional to $k/m$ (where $m$ is the vertical wavenumber), as from the continuity equation, while for Kelvin waves, $k$ and $m$ do not have
inter-dependence in their dispersion relationship.

     For the antisymmetric mode, $F_z$ has similar spectral shapes and roughly the same $(k, \omega)$ peaks as those of $v_{\mathrm{S}}$ for each reanalysis below $20\,\mathrm{hPa}$ (Figs. 2 and 10). Consistent with the $v_{\mathrm{S}}$ spectra, the frequency bounds of the $F_z$ spectra at $70\,\mathrm{hPa}$ are quite a bit lower in JRA-55, JRA-55C, and MERRA-2 than in the other reanalyses. At $20\,\mathrm{hPa}$, the $F_z$ spectra differ from the $v_{\mathrm{S}}$ spectra in that the relatively low frequency and high wavenumber (and thus low phase speed) components are largely suppressed
in the $F_z$ spectra. The major term of $F_z$ for MRG waves is the meridional heat flux ($v'T'$), and we confirm that antisymmetric temperature perturbations tend to be small in that low-frequency range (not shown). The vertical group velocity of MRG waves depends largely on their frequency (Andrews et al., 1987): i.e., low-frequency waves have smaller group velocities. This may result in slower vertical propagation and apparently stronger dissipation of the relatively low-frequency MRG waves with an increase in altitude compared to the high-frequency components. The linear solution of free MRG waves also indicates that





among variables, the temperature amplitude is proportional to the frequency (Andrews et al., 1987). This may explain the observation that the dependence of MRG wave amplitudes upon the frequency could be reflected more in the temperature spectra than in the $v_S$ spectra.

The meridional EP flux spectra are also calculated in the same way to obtain the spectra of EP flux divergence (not shown). Then, the EP flux divergence and $F_z$ spectra are reconstructed as a function of phase speed to investigate the vertical propagation and dissipation of Kelvin and MRG waves during different QBO phases and to quantify the QBO forcing by those waves as well as the phase-speed ranges responsible for the forcing. For this, we construct bins of phase speeds ($c_j$) with a width of $2\,\mathrm{m\,s^{-1}}$ ($\Delta c$) and integrate the $k$–$\omega$ spectral densities of EP flux divergence and $F_z$ across the corresponding bins (i.e., $(2\pi a)\omega/k \in [c_j - \Delta c/2, c_j + \Delta c/2]$, where $a$ is the mean radius of the Earth).[1] For each wave type, the same filters as described in Section 3.1 are used: $h = 8$–$240\,\mathrm{m}$, $k = 1$–$10$, and $0.05 < \omega \le 0.5\,\mathrm{cyc\,day^{-1}}$ for Kelvin waves and $h > 8\,\mathrm{m}$, $|k| = 1$–$10$, and $\omega < 0.33\,\mathrm{cyc\,day^{-1}}$ for MRG waves.

Figure 11 shows vertical profiles of $15°\mathrm{N}$–$15°\mathrm{S}$ averaged EP flux divergence and $F_z$ as a function of phase speed, for Kelvin and MRG waves at $c > 0$ and $c < 0$, respectively, composited for four different phases of the QBO. The composite is made by selecting one month for each QBO cycle when the zonal wind tendency is largest at a given altitude and then averaging over the 13 cycles available. The observed monthly-mean near-equator wind profiles compiled by the Freie Universität Berlin (FUB) are used to select the months. The four composites presented in Fig. 11 are for the months of maximum westerly and easterly tendencies at 20 and $50\,\mathrm{hPa}$. The results are generated using the ML datasets of the four reanalyses available for EP flux calculation (ERA-I, MERRA-2, JRA-55, and JRA-55C), since the EP flux could be largely underestimated when using the SL datasets. Furthermore, the height dependence of the amplitude damping, as reported in Table 3, could also affect the estimation of the vertical divergence of EP flux (see Fig. 3 in Kim and Chun, 2015, for a large difference in the EP flux divergence between ML and SL datasets of ERA-I).

The EP flux divergence of the Kelvin waves is found to be quite different between reanalyses not only in magnitude but also in its peak altitude. At the maximum westerly tendency phase at $20\,\mathrm{hPa}$ (Fig. 11, the first row), the EP flux and its divergence are notably the largest in ERA-I up to $\sim$15 hPa. From $\sim$15 hPa upwards, JRA-55 has the largest flux, resulting in a higher altitude of the Kelvin wave forcing (peak at $\sim$12 hPa) than ERA-I (15 hPa) and MERRA-2 (18 hPa). Kawatani et al. (2016) showed that the easterly-to-westerly transitions of the QBO at $10\,\mathrm{hPa}$ are faster (slower) in MERRA-2 (JRA-55) than in ERA-I (see their Fig. 16), which implies lower (higher) altitudes of the westerly shear layers below $10\,\mathrm{hPa}$. The difference in these shear altitudes is likely to be at least part of the cause of the different forcing altitudes in Fig. 11. The phase speed at which the forcing occurs increases with altitudes from $\sim$10–30 m s$^{-1}$ at 20 hPa to 15–40 m s$^{-1}$ at 10 hPa. For the $15°\mathrm{N}$–$15°\mathrm{S}$ average, the Kelvin wave forcing integrated for the phase speed has maxima of 4.7, 3.8, and 4.1 m s$^{-1}$ month$^{-1}$ in ERA-I, MERRA-2, and JRA-55, respectively. When the westerly tendency is maximized at $50\,\mathrm{hPa}$ (Fig. 11, the second row), the Kelvin wave

---

[1]Prior to this procedure, we converted the $k$–$\omega$ spectra to much finer resolution in frequency using linear interpolation for each $k$. This is required because the frequency resolution of the original spectra ($\Delta\omega = 1/90\,\mathrm{cyc\,day^{-1}}$) is coarse in terms of $c$ for small $k$ (e.g., for $k = 1$, $(2\pi a)\Delta\omega/k \sim 5\,\mathrm{m\,s^{-1}} > \Delta c$), causing artificial peaks and noise. For the finer spectra, we set $\Delta\omega$ to be 500 times smaller than the original, after confirming that the results converge without displaying artificial peaks for varying $\Delta\omega$ around such values.





forcing peaks at approximately $40\,\mathrm{hPa}$ in all of the reanalyses and has a maximum of $3.8\,\mathrm{m\,s^{-1}\,month^{-1}}$ in ERA-I. This reanalysis exhibits the largest flux at all phase speeds below $40\,\mathrm{hPa}$, while at $30\,\mathrm{hPa}$ the flux and forcing are comparable in ERA-I and JRA-55. At the other two phases of the QBO (Fig. 11, the third and last rows), the zonal wind is westerly at $70\,\mathrm{hPa}$, causing less upward propagation of Kelvin waves in the lowermost stratosphere.

MRG waves dissipate largely in the lower stratosphere when the zonal wind is easterly at $70\,\mathrm{hPa}$ (Fig. 11, the first and second rows). The MRG wave forcing is largest in these phases, although its magnitude is only up to $\sim 1\,\mathrm{m\,s^{-1}\,month^{-1}}$ in all of the reanalyses. When the easterly tendency is maximum at $20\,\mathrm{hPa}$ (Fig. 11, the third row), MRG waves propagate through the lower stratospheric westerlies and dissipate at around $30$–$20\,\mathrm{hPa}$. The forcing by the MRG waves at these altitudes is largest in MERRA-2 ($\sim 0.5\,\mathrm{m\,s^{-1}\,month^{-1}}$), where it occurs at phase speeds between $-10$ and $-35\,\mathrm{m\,s^{-1}}$.

As mentioned in the Introduction, the momentum budget of the QBO is currently not fully constrained by observations, and it is believed to be essential for modeling the QBO. The different altitudes of Kelvin wave forcing around $15\,\mathrm{hPa}$ among the reanalyses, as well as the different magnitudes, shown in Fig. 11 may imply one limitation in quantifying wave forcing of the QBO using a reanalysis dataset. One would expect that if forcing at different altitudes is mimicked in a simple QBO model (e.g., Plumb, 1977), it may simulate a QBO that has rather different characteristics. Advances in observation and assimilation of stratospheric waves might be needed to reduce the spread of the assimilated waves and to further improve global models. The rather weak magnitudes of the MRG wave forcing in all of the reanalyses imply that other waves (especially gravity waves) might play a more important role in driving the easterly phase of the QBO than MRG waves (Kawatani et al., 2010; Evan et al., 2012; Ern et al., 2014) and/or that current observational measurements, as well as models and assimilation methods, cannot fully capture the MRG waves which have relatively small vertical wavelengths. Modeling studies have demonstrated that adequate representation of the stratospheric equatorial waves requires vertical resolutions of $\sim 700\,\mathrm{m}$ or finer in the lower to middle stratosphere (Giorgetta et al., 2006; Richter et al., 2014), while the vertical spacings of the prediction models used for the current reanalyses are about a factor of two larger than that (Supplement to Fujiwara et al., 2017).

## 4    Further investigation of the high-frequency disturbances in the middle stratosphere

In the $k$–$\omega$ spectrum for the symmetric component of meridional wind (Fig. 2), a distinct spectral peak is seen at $20\,\mathrm{hPa}$,
which stretches toward high frequencies and high wavenumbers from around $\omega \sim 0.4\,\mathrm{cyc\,day^{-1}}$ and $k \sim -3$. To investigate this further, we filter a part of the spectrum that does not coincide with the spectrum of the lower-frequency MRG waves investigated in Section 3. The filtered spectral region is $0.5 < \omega < 0.7\,\mathrm{cyc\,day^{-1}}$ for $k = -7$ and $0.6 < \omega < 0.75\,\mathrm{cyc\,day^{-1}}$ for $k = -8$. Figure 12 presents time series of variances of the filtered meridional-wind spectrum at $20$ and $10\,\mathrm{hPa}$, along with the observed near-equator zonal wind at each altitude compiled by FUB. For all reanalyses, it is clearly seen that this spectral
peak appears only when the background wind is easterly with substantial speeds at each altitude. Furthermore, there exist time lags of the variance peaks between the two altitudes, as well as a lag in the zonal wind due to the downward progression of the QBO. Due to these time lags, there exist periods during which the variance is large at $10\,\mathrm{hPa}$ while it is much smaller at $20\,\mathrm{hPa}$ at the same time (e.g., late 1999/early 2000 and late 2008/early 2009). This indicates that the spectral signal of these



waves does not originate from below. On the other hand, there also exist periods during which the variance is much larger at 20 hPa than at 10 hPa (e.g., late 2007), suggesting that the waves at 20 hPa are, at least in part, generated in situ. In fact, the vertical EP flux of this spectrum is not well identified (see Fig. 10), which could indicate a minimal preference for upward or downward propagation of the waves. This also could be suggestive of in-situ wave generation. This spectral peak is also seen

in the upper stratosphere above 10 hPa, although it is less evident here because the peak is not clearly separated from that of the lower-frequency waves (not shown). It is reminiscent of the MRG waves reported by Maury and Lott (2014), which were found at $\sim$20 hPa when the easterly jet exists below.

Given the dependence of that spectrum filtered for $k = -7$ and $-8$ upon the background wind $U$ (Fig. 12), the entire spectrum of those waves can be more clearly identified by compositing for the periods with strong background easterlies.

Figure 13 presents two composites of the symmetric meridional-wind spectrum for the months with $U < -25\,\mathrm{m\,s^{-1}}$ and $U > -20\,\mathrm{m\,s^{-1}}$ at 20 hPa, based on the FUB zonal wind. The composite result is shown only for JRA-55C (in which the amplitude of this spectrum is the largest in Fig. 2) but is qualitatively similar for the other reanalyses. For $U < -25\,\mathrm{m\,s^{-1}}$, it is shown that the spectrum has peaks at $k = -4$ and $-5$ with periods of about 2 days. The ground-based zonal phase speeds corresponding to the peaks are between $-45$ and $-60\,\mathrm{m\,s^{-1}}$. These are much higher compared to those of the waves detected

in Maury and Lott (2014) (about $-19\,\mathrm{m\,s^{-1}}$). The spectral region is best identified by the dispersion curves of MRG waves (Fig. 13) rather than the other types of waves, provided that $U \sim -30\,\mathrm{m\,s^{-1}}$. The spectrum composited for $U > -20\,\mathrm{m\,s^{-1}}$ is similar to that of the lower-frequency MRG waves observed in the lower stratosphere. The variances of the symmetric meridional wind by the waves observed at 20–10 hPa during the easterly QBO phase typically have magnitudes of roughly half of those by the lower-frequency MRG waves during the westerly phase (not shown). More details on the structure of the

waves in a strong easterly background wind and the possible impacts of these waves in the tropical stratosphere will be pursued in a future study.

## 5   Summary

The equatorial Kelvin and MRG waves in the TTL and stratosphere represented in six reanalyses (ERA-I, MERRA, MERRA-2, CFSR, JRA-55, and JRA-55C) are compared for the period of 1981–2010. The power spectra with respect to the zonal

wavenumber and frequency are presented (Figs. 1 and 2). The spectral shapes of the Kelvin and MRG waves are broadly similar among the reanalyses: they exhibit common spectral peaks and widths as well as vertical variations of the spectral shapes, except for the Kelvin waves above 100 hPa in CFSR. The stratospheric Kelvin waves in CFSR have remarkably larger powers at relatively low-frequency, low-equivalent-depth ranges than in the others. JRA-55 and JRA-55C show relatively smaller temperature amplitudes than the other reanalyses, common to all altitudes below 20 hPa.

The spatial distributions and patterns of the equatorial waves are investigated (Figs. 3–5). It is shown in all of the reanalyses that Kelvin and MRG wave variances tend to be large in the eastern and western hemispheres, respectively, and that the locations of their maxima tilt eastward in the vertical. However, the longitudinal variations of the Kelvin wave variances are much smaller in CFSR compared to those in the others. The leading mode EOFs of the Kelvin- and MRG-wave filtered perturbations are used




to obtain statistically representative patterns of the waves in each reanalysis, following Kiladis et al. (2016). For both waves, the horizontal winds and geopotential perturbations projected onto the leading EOFs show spatial patterns and polarization relationships consistent with the classical equatorial wave theory. All the reanalyses studied here exhibit remarkably similar patterns for the Kelvin wave leading modes, which have zonal wavenumber 1 structures with larger amplitudes in the eastern hemisphere. The MRG wave leading modes are confined over the eastern Pacific to Atlantic sector and also show reasonable agreement between the reanalyses, although their zonal scales are somewhat larger in CFSR.

From analysis of the time series of Kelvin and MRG wave variances, systematic differences are found between the periods before and after the late 1990s in several aspects (Figs. 6–8).

– The difference in the Kelvin wave variances between JRA-55 and JRA-55C (which stems from the exclusion of satellite data in the assimilation in JRA-55C) shows significant changes after the late 1990s at $100\,\mathrm{hPa}$ (from $\sim$7% to 10–24%) and at 10 and $5\,\mathrm{hPa}$ (from between 0 and $-10\%$ to 10–20%).

– The Kelvin wave variances in the middle and upper stratosphere in ERA-I exhibit a large increase around 1998 ($\sim$50%), becoming significantly larger than those in the other reanalyses (Figs. 8 and S3).

– In the middle and upper stratosphere in MERRA-2, the peaks of the Kelvin wave variances during westerly-shear phases of the QBO are not represented in some years before the mid-1990s (Fig. S3). The MRG wave variances at $10\,\mathrm{hPa}$ show exceptionally large values during the westerly-shear phases until 1998, but not afterward.

– The MRG wave variances in CFSR fluctuate largely in the vertical from $\sim$30 hPa before 1998, but not afterward.

The results listed here demonstrate significant impacts of the TOVS–ATOVS transition starting in 1998 on the assimilated wave amplitudes in the middle and upper stratosphere in all the reanalyses. Below $10\,\mathrm{hPa}$, the satellite impacts on the waves are identifiable in JRA-55 only by direct comparison with JRA-55C at $100\,\mathrm{hPa}$ where the SSU instruments in the TOVS suite do not cover.

The time series of the Kelvin wave variances at $100\,\mathrm{hPa}$ (Fig. 6) exhibit increasing trends in all the reanalyses except JRA-55C. However, the veracity of the trends in Kelvin wave activity is uncertain due to potential satellite effects in the reanalyses. On the other hand, the MRG wave variances show a long-term increase at $100\,\mathrm{hPa}$, common to all of the reanalyses including JRA-55C which is not affected by satellite transitions.

It is also noted that variances of equatorial waves can be underestimated by up to around 30% when standard-pressure-level datasets are used. The wave perturbations in these datasets are damped out by vertical interpolation, and the damping effect is large for waves with small vertical wavelengths.

The EP flux and its divergence are presented as a function of phase speed and height to compare the QBO forcing measured in the reanalyses (Fig. 11). In general, the Kelvin wave forcing is the largest in ERA-I, while the phase speeds at which the forcing occurs are comparable among ERA-I, MERRA-2, JRA-55, and JRA-55C. However, for the QBO phases with westerly acceleration at $\sim$15 hPa, the height of the peak forcing differs between the reanalyses by up to $\sim$3 km. The MRG wave forcing is in general small in all the reanalyses (up to $\sim$1 m s$^{-1}$ month$^{-1}$ for the 15°N–15°S average).





In addition, waves detected in the spectra of symmetric meridional winds at $20\,\mathrm{hPa}$ (Fig. 2), which are distinguishable from the MRG waves observed below, are further investigated (Figs. 12 and 13). It is found in all the reanalyses that they appear only when the background wind is easterly, unlike the MRG waves propagating from below, with ground-based periods of about 1–3 days. Their spectral shape is best identified by the dispersion curve of MRG waves with the background wind of

5 about $-30\,\mathrm{m\,s^{-1}}$. More details regarding the spatial pattern, origin, and implication of these waves remain to be studied in the future.

*Author contributions.* The EOF analysis in Section 3.2 was carried out by GNK, JRA, and JD, and the other calculations were by YHK. The initial idea for the paper was provided by MF and JA. All authors provided further ideas and contributed to the interpretation of the results. The manuscript was written first by YHK with advice by CY, except Section 3.2 written by JRA and GNK. All authors contributed

to improvement/correction of the manuscript.

*Competing interests.* The authors declare that they have no conflict of interest.

*Special issue statement.* This article is part of the special issue "The SPARC Reanalysis Intercomparison Project (S-RIP) (ACP/ESSD inter-journal SI)". It is not associated with a conference.

*Acknowledgements.* YHK and CY were supported by the National Research Foundation of Korea (NRF-2016R1C1B2006310). MF was

15 financially supported in part by the Japan Society for the Promotion of Science (JSPS) through Grants-in-Aid for Scientific Research (26287117, 16K05548, and 18H01286). CW is funded by a Royal Society University Research Fellowship (ref. UF160545) and by Natural Environment Research Council grant NE/R001391/1.




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



**Table 1.** Reanalyses used in this study.

| Abbreviation | Full name | Reference |
|---|---|---|
| ERA-I | European Centre for Medium-Range Weather Forecasts Interim Reanalysis | Dee et al. (2011) |
| MERRA | Modern-Era Retrospective Analysis for Research and Applications | Rienecker et al. (2011) |
| MERRA-2 | Modern-Era Retrospective Analysis for Research and Applications, Version 2 | Gelaro et al. (2017) |
| CFSR | Climate Forecast System Reanalysis | Saha et al. (2010) |
| JRA-55 | Japanese 55-year Reanalysis | Kobayashi et al. (2015) |
| JRA-55C | Japanese 55-year Reanalysis assimilating conventional observations only | Kobayashi et al. (2014) |

MERRA and MERRA-2 each provide two sets of products called ANA (analysis state) and ASM (assimilated state) (see Bloom et al., 1996, for the details),
and the latter is used here.





**Table 2.** Percentage of variance explained by the leading EOF pairs representing Kelvin and MRG waves for each reanalysis.

|         | Kelvin | MRG  |
| ------- | ------ | ---- |
| ERA-I   | 41.5   | 19.1 |
| MERRA   | 35.8   | 19.4 |
| MERRA-2 | 36.3   | 19.7 |
| CFSR    | 32.0   | 18.2 |
| JRA-55  | 35.4   | 19.0 |
| JRA-55C | 37.6   | 17.9 |



**Table 3.** Differences in the mean variances over 1981–2010 between the ML and SL results, relative to the ML results (%), for the (left) Kelvin wave temperature and (right) MRG wave meridional wind.

|  | ERA-I | MERRA-2 | CFSR | JRA-55 | JRA-55C |
|---|---|---|---|---|---|
| 5 hPa | 10 / 8 | 19 / 14 | 7 / 8 | 13 / 1 | 12 / 1 |
| 7 hPa | 16 / 15 | 4 / 3 | 23 / 38 | 13 / 1 | 12 / 1 |
| 10 hPa | 3 / 4 | 9 / 10 | 13 / 26 | 13 / 0 | 12 / 0 |
| 20 hPa | 15 / 22 | 17 / 27 | 8 / 21 | 13 / 3 | 13 / 4 |
| 30 hPa | 13 / 19 | 11 / 18 | 13 / 22 | 16 / 2 | 16 / 2 |
| 50 hPa | 29 / 27 | 16 / 18 | 16 / 20 | 20 / 3 | 20 / 3 |
| 70 hPa | 25 / 26 | 18 / 16 | 17 / 21 | 23 / 4 | 22 / 3 |
| 100 hPa | 15 / 13 | 4 / 3 | 16 / 15 | 19 / 0 | 19 / 0 |





**Figure 1.** Zonal wavenumber–frequency power spectra of the symmetric component of temperature at 100, 70, 50, and 20 hPa (from bottom to top) from the standard-level datasets of six reanalyses (ERA-I, MERRA, MERRA-2, CFSR, JRA-55, and JRA55-C: from left to right), averaged over 15°N–15°S in 1981–2010. The power spectra are presented in the variance-preserving form with log-scale axes. The Kelvin wave dispersion curves are indicated by black solid for the equivalent depths ($h$) of 8, 60, and 240 m. The ratio of the spectral power to that of the background spectrum is indicated by thin purple for the values of 1.5, 2, 3, and 5.





**Figure 2.** The same as in Fig. 1 except for the spectra of the symmetric component of meridional wind. The mixed Rossby-gravity (MRG) wave dispersion curves for the windless background state are indicated by dotted for $h = 8$, 60, and 480 m. At 100 hPa, the dispersion curves for the background zonal wind of $+10\,\mathrm{m\,s^{-1}}$ are also indicated (dashed).





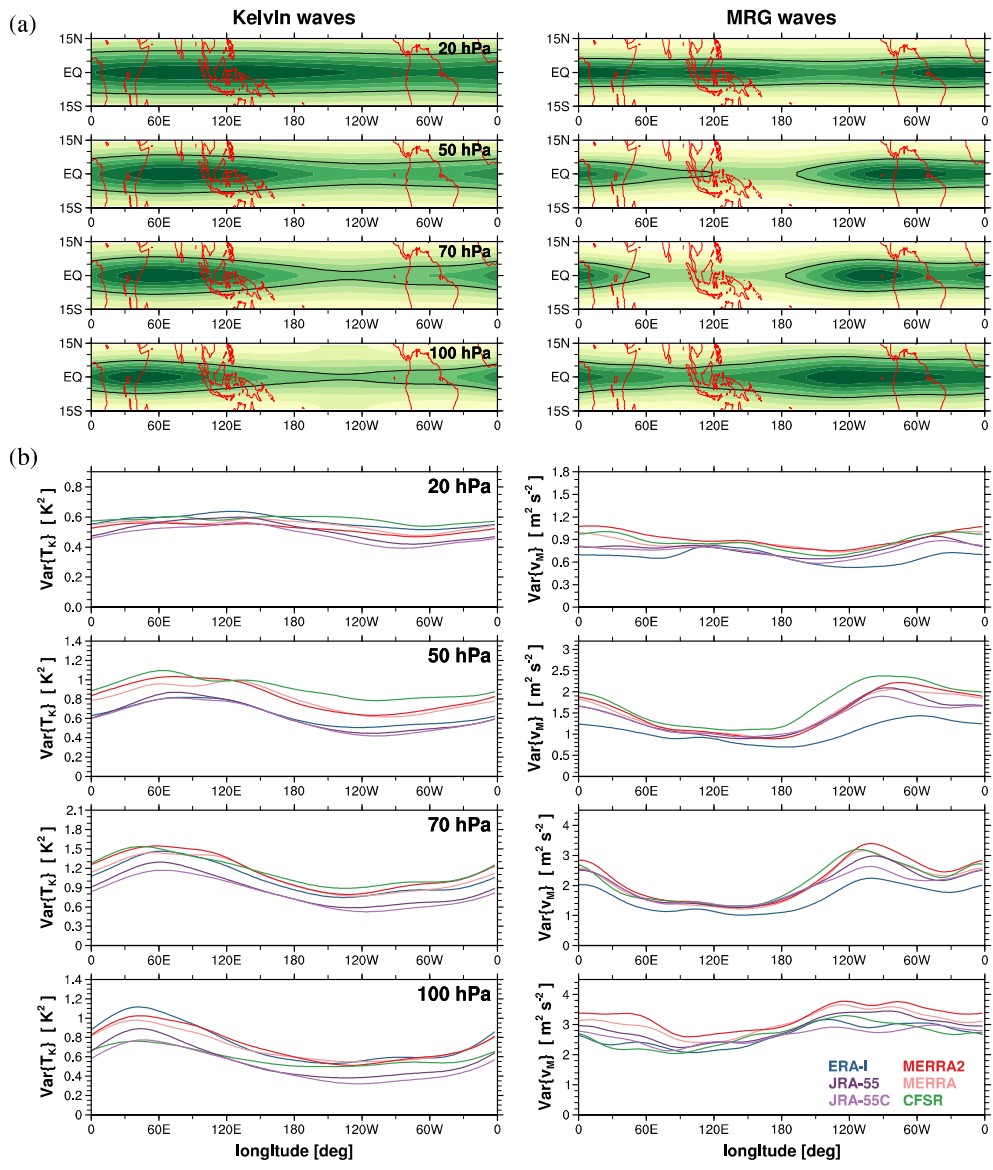

**Figure 3.** Horizontal distributions of variances of (left) temperature for the Kelvin waves ($T_{\mathrm{Kelvin}}$) and (right) meridional wind for the MRG waves ($v_{\mathrm{MRG}}$) at 100, 70, 50, and 20 hPa averaged for 1981–2010: (a) ensemble mean for ERA-I, MERRA-2, CFSR, and JRA-55, and (b) distributions at the equator for each of the six reanalyses. In (a), the mean variances are normalized by their maximum value on each horizontal plane. The shading interval is 0.1 with the black contour indicating 0.5. See the text in Section 3.1 for the definitions of $T_{\mathrm{Kelvin}}$ and $v_{\mathrm{MRG}}$.

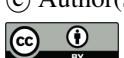



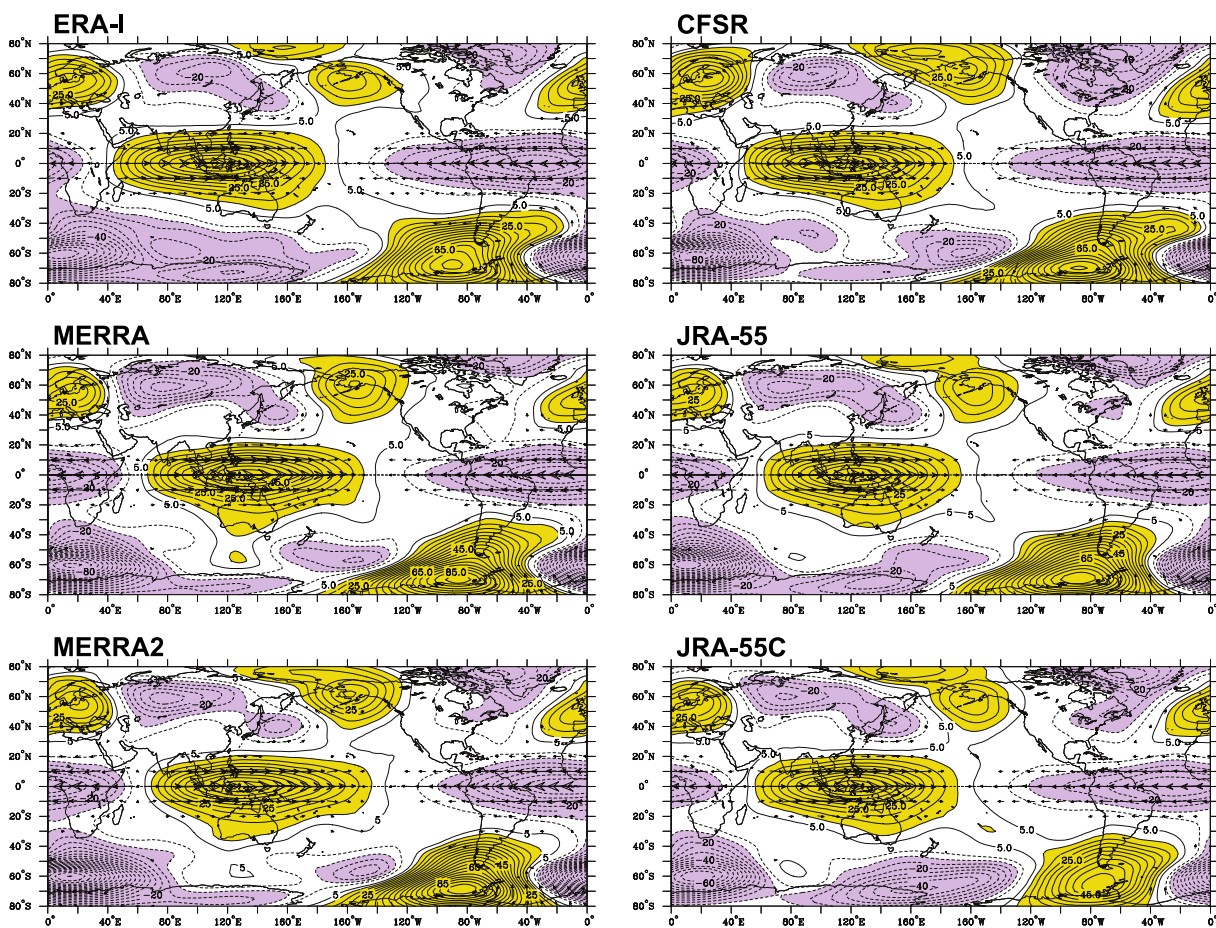

**Figure 4.** Horizontal wind (arrow) and geopotential perturbations (shading) projected onto the principle component times series of the first EOF modes for Kelvin waves at 50 hPa in each reanalysis. The EOFs for Kelvin waves are calculated using 2–25 day filtered eastward-propagating zonal winds (see the text). The winds are shown at the locations where they are statistically significant.



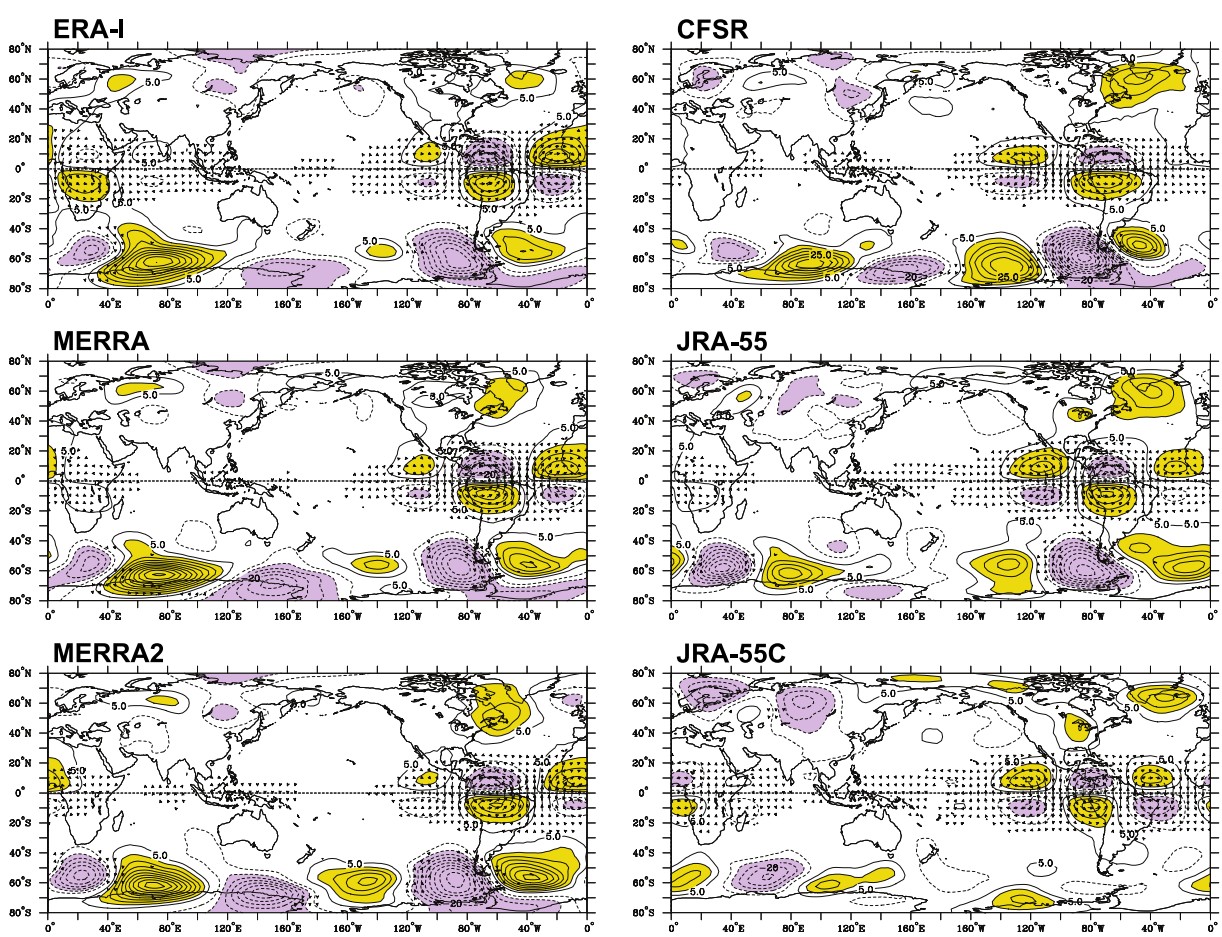

**Figure 5.** The same as in Fig. 4 except for MRG waves. The EOFs for MRG waves are calculated using 2–6 day filtered meridional winds.





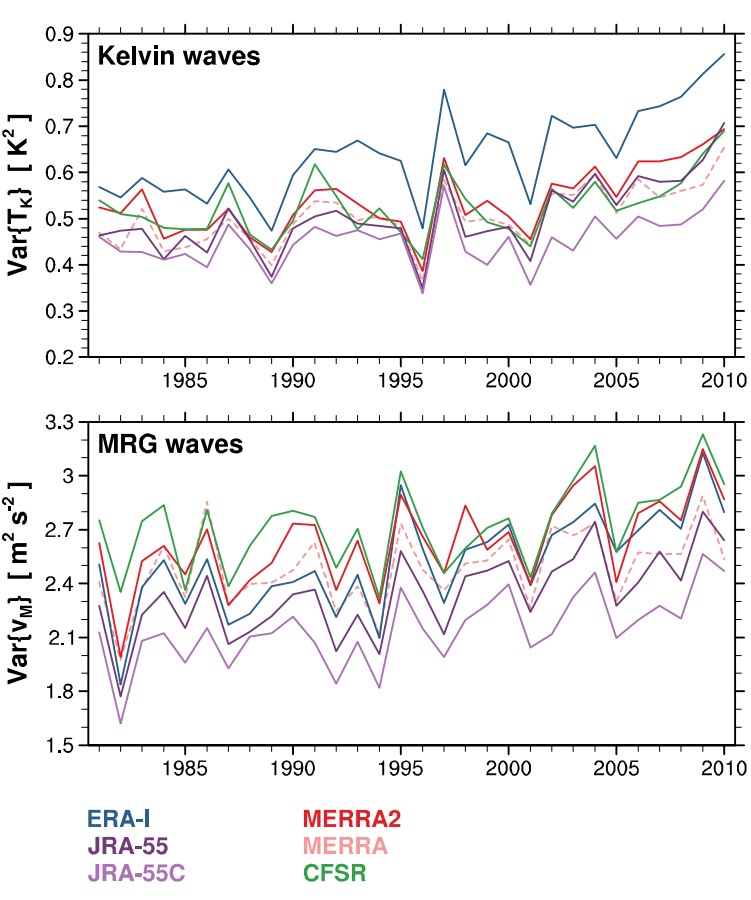

**Figure 6.** Annual-mean time series of variances of (upper) $T_{\mathrm{Kelvin}}$ and (lower) $v_{\mathrm{MRG}}$ at 100 hPa averaged over 15°N–15°S. The MERRA results are from the standard-level datasets (dashed), and the others from the model-level datasets (solid).



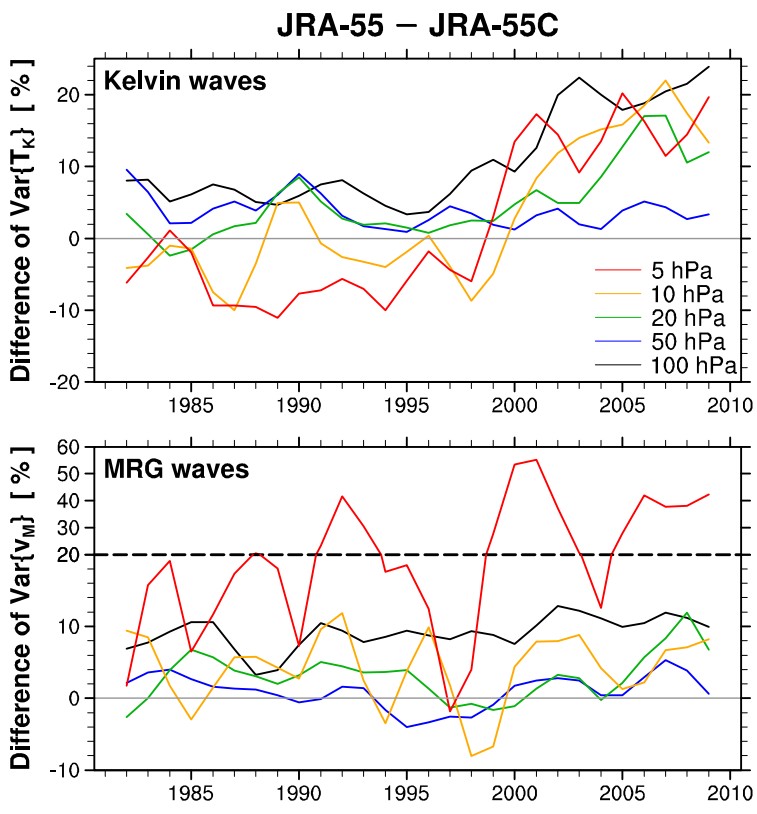

**Figure 7.** Differences in the annual-mean variances of (upper) $T_{\mathrm{Kelvin}}$ and (lower) $v_{\mathrm{MRG}}$ over 15°N–15°S between JRA-55 and JRA-55C, relative to the 30-year mean variances in JRA-55C, at various altitudes. The 1-2-1 smoothing is applied to the differences to filter out the interannual fluctuations by the QBO. The results are obtained from the model-level datasets.



**Figure 8.** Variance profiles of (upper) $T_{\text{Kelvin}}$ and (lower) $v_{\text{MRG}}$ averaged over the periods of (left) 1981–1997 and (center) 1999–2010, and (right) their differences (1999–2010 minus 1981–1997) relative to the 30-year average (1981–2010). The dashed and solid indicate the results from the standard-level and model-level datasets, respectively.





**Figure 9.** Zonal wavenumber–frequency spectra of the vertical EP flux, multiplied by $-1$ $(-F_z)$, for symmetric modes at 100, 70, 50, and 20 hPa (from bottom to top) from the standard-level datasets of six reanalyses (ERA-I, MERRA, MERRA-2, CFSR, JRA-55, and JRA55-C: from left to right), averaged over $15°$N–$15°$S in 1981–2010. The Kelvin wave dispersion curves are indicated by black solid for $h = 8$, 60, and 240 m.



**Figure 10.** The same as in Fig. 9 but for $F_z$ of anti-symmetric modes. The MRG wave dispersion curves for the windless background state are indicated by dotted for $h = 8$, 60, and 480 m. At 100 hPa, the dispersion curves for the background zonal wind of $+10\,\mathrm{m\,s^{-1}}$ are also indicated (dashed).



**Figure 11.** Vertical profiles of phase-speed spectra of the EP flux divergence (shading) and vertical EP flux (black contour) for the Kelvin waves at $c > 0$ and MRG waves at $c < 0$, composited for the QBO phases of maximum westerly tendencies at 20 and 50 hPa (the first and second rows, respectively) and easterly tendencies at 20 and 50 hPa (the third and last rows, respectively), from the model-level datasets of ERA-I, MERRA-2, JRA-55 and JRA-55C. The zonal wind profiles for those composites are also indicated (green; see the text for the composite method). The contour intervals of the EP flux are 0.005 and 0.0005 mPa / m s$^{-1}$ for the Kelvin and MRG waves, respectively, and every third contour is distinguished by thicker lines.



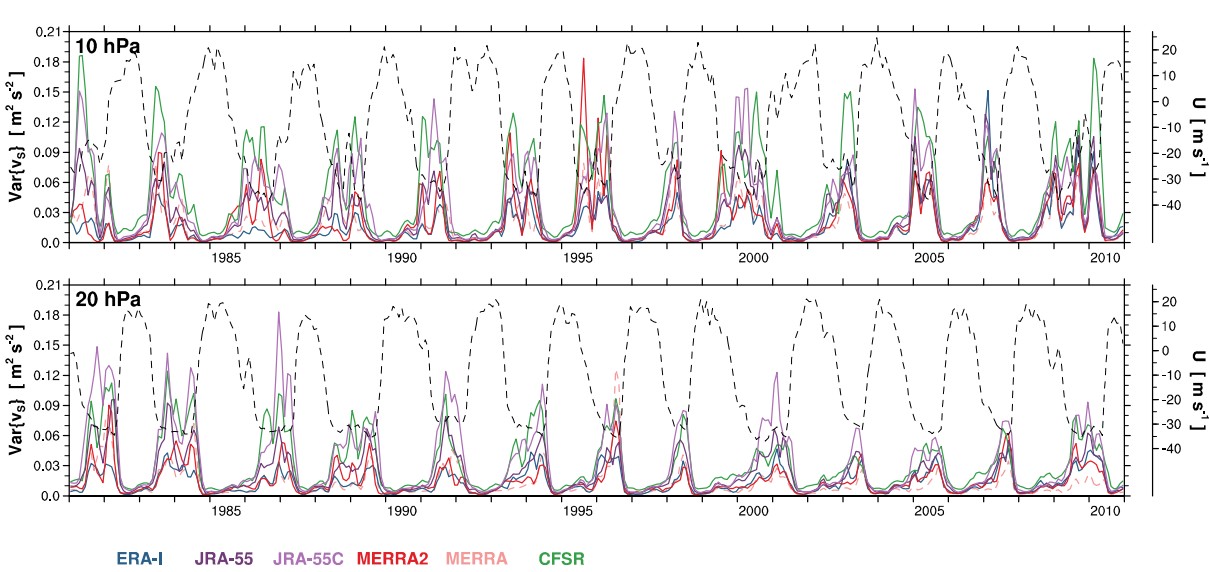

**Figure 12.** Monthly time series of variances of the symmetric component of meridional wind at (upper) 10 and (lower) 20 hPa, filtered for $0.5 < \omega < 0.7 \, \mathrm{cyc \, day^{-1}}$ with $k = -7$ and $0.6 < \omega < 0.75 \, \mathrm{cyc \, day^{-1}}$ with $k = -8$ (see Fig. 2). The monthly zonal wind from radiosonde observations is also presented at each level (dashed black).



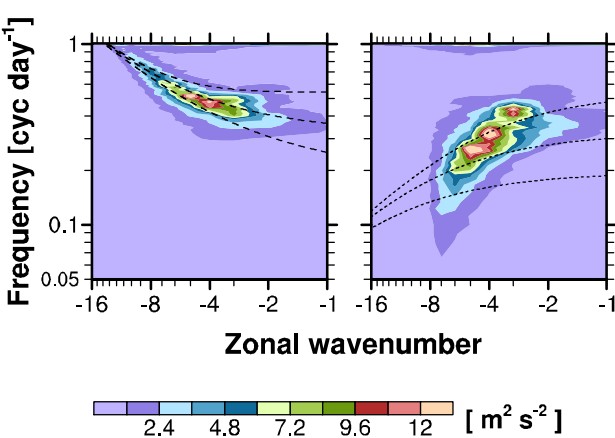

**Figure 13.** The same as in Fig. 2 (rightmost in the first row) except for the average over the months when (left) the zonal wind $U < -25\,\mathrm{m\,s^{-1}}$ and (right) $U > -20\,\mathrm{m\,s^{-1}}$, at $20\,\mathrm{hPa}$ in JRA-55C. The dashed in the left panel indicate the MRG wave dispersion curves for $h = 8$, 60, and $480\,\mathrm{m}$ for the background wind of $-30\,\mathrm{m\,s^{-1}}$, and the dotted in the right indicate those for the windless background state.