# Peer review of "Comparison of equatorial wave activity in the tropical tropopause layer and stratosphere represented in reanalyses"

_Atmospheric Chemistry and Physics, 2019_

## Referee Comment (RC1) · Anonymous Referee #1 · 11 Mar 2019

This paper presents analysis of stratospheric equatorial waves in several reanalysis data sets, focusing on standard space-time spectral analysis of Kelvin and Mixed Rossby Gravity waves. The analyses are mostly straightforward and the results show reasonable agreement among the reanalyses (and with previous publications), with a few outliers identified. Long-term variations in wave variances show changes likely related to input satellite data sets, in particular the transition from TOVS to ATOVS in 1998. Comparisons are also made for derived EP fluxes for the Kelvin and MRG waves. The overall results provide quantitative information on tropical wave behavior among the reanalyses, and the paper makes an original and useful contribution to the SRIP evaluations. The paper is reasonably well written and is appropriate for ACP. I

have a number of mostly minor comments for the authors to consider in revision.

1) The inclusion of JRA55C (no satellite data) is especially nice for quantifying the influence of satellite data in the reanalyses, and I'm surprised at how small the differences are with JRA55 at upper levels (where radiosonde data are sparse). How should this result be interpreted, i.e. are these wave spectra mostly characteristic of the forecast model, or are upper levels constrained by lower levels?

2) Figure 11 uses the Singapore zonal winds as a standard reference for all reanalyses. Are there any systematic differences found using the zonal winds from the separate reanalyses instead of a single reference? Does behavior of the equatorial waves in the different reanalyses 'feel' the differences in zonal winds identified in Kawatani et al (2016)?

3) Can you please add a right hand axis indicating period (in days) for the various spectra (Fig. 1, 2, 9,10,13). The discussions in the text are all based on wave period, and it would be helpful to simply see this on the figures.

4) Are there any systematic seasonal variations in the Kelvin and MRG spectra at lower levels (100 hPa) identified in the reanalyses? Since you have long records of monthly statistics it would be easy to identify such behavior.

5) The EP flux calculations are complicated and it would be helpful to have a direct reference to the calculations used here (beyond the standard Andrews 1987 textbook). Are these calculations identical to those used in Kim and Chun (2015)? How do the climatological EP fluxes here compare to other published results?

6) The composited phase-speed spectra results are very nice (Fig. 11). For the conversion from frequency to phase-speed there is a factor of zonal wavenumber needed to conserve spectral density (see Randel and Held, 1991, JAS). Has this been incorporated in these calculations? While the composite results are revealing, it would be interesting to see the variability of some of these diagnostics within the composites, for

example perhaps showing a 'spaghetti plot' of the separate composited time series to see the actual variability, within and among the reanalyses. Could this be included as a Supplementary figure?

7) A recent paper on observed lower stratosphere Kelvin waves is Scherrlin-Pirscher et al 2017 doi:10.5194/acp-2016-576, and this might be a useful reference to include.

8) The high frequency disturbance in the meridional wind at upper levels has some characteristics similar to the so-called 2-day wave (zonal waves 3-4, ~2 day period, occurrence in strong easterly winds); see e.g. https://doi.org/10.1029/2009JD012239. It might be useful to look at the latitude-height structure of the waves for a tentative identification.

---

## Referee Comment (RC2) · Anonymous Referee #2 · 13 Mar 2019

A very clear presentation of a thorough analysis and comparison of equatorial waves in various reanalyses. In Fig. 8, I would recommend that the same x-axis is used for the difference in T as in v.

---

## Referee Comment (RC3) · Rolando R. Garcia (Referee) · 15 Mar 2019

Review of "Comparison of equatorial wave activity in the tropical tropopause layer and stratosphere represented in reanalyses"

by Y.-H. Kim et al.

Recommendation: minor revision

This is a well-written, comprehensive comparison of Kelvin and Rossby-gravity waves and wave activity as represented in six reanalysis datasets. The comparison is thorough and the interpretation of the results is reasonable, as far as it goes. However, I

believe there are two areas where the paper could be improved. The first (required) is the choice of latitude range over which results are averaged for Eliassen-Palm (EP) flux and EP flux divergence comparisons. Garcia and Richter (JAS, 2019) have recently shown that averaging beyond $\pm 5°$ can be misleading in the case of Rossby-gravity waves because their EP flux divergence pattern changes sing within a narrow neighborhood of the Equator, such that broader latitude averaging leads to cancellation. The second (optional) would be a more thorough examination of the impact of the quasi-biennial oscillation (QBO) on the behavior of EP fluxes and wave spectra (at present, there is only one figure—Fig. 11—and a short discussion thereof).

Otherwise, the paper is an important contribution to the literature on tropical waves, and includes a very useful discussion of the impact of new satellite observations on the reanalysis products. I believe the paper is suitable for publication once the general comments above and the specific comments listed below are addressed.

Specific comments (page, line):

(4, 15) "the EP flux formulation": The reference cited does not explain how the flux is calculated; it just gives the standard definition of EP flux. A brief description of how you go from spectral components of velocity and temperature to F(omega,k) would be helpful. Also, do you average F in latitude? Over what range? See also comment at (10,14).

(5,4) "JRA-55 and JRA-55C show . . . less power below 20 hPa": Does this have anything to do with vertical resolution? Slower Kelvin waves would be prevalent in the lower stratosphere; these waves have short vertical wavelengths whose accurate representation depends on having sufficient vertical resolution. It would be useful to include in Table 1 information on the horizontal and vertical resolution of each reanalysis.

(5,6) "thin purple" Thin purple what? Are you referring to the thin purple lines in the figure?

(5,15) "MRG generated in the region. . .": How do you know where the waves are generated?

(5,17) "Fig. 2, dashed": Figure 2 has many dashed lines. Do you mean the longer-dashed lines in the panels for 100 hPa?

(5,22) "more intense than those at lower frequencies with |k| > 4, as the altitude increases": I am not sure what this means. There are local maxima at the (omega,k) mentioned in the text at 50 and 20 hPa. At 20 hPa, these maxima are larger than any other spectral components, although this is not the case at 50 hPa. Is that what you have in mind? I am not sure why the remark about |k|>4 is needed here.

(6,14) "MRG . . . wavepacket travels eastward": While this is evident from the zero-background wind dispersion relation, it may not be obvious to many readers, who are conditioned to think of RG wavepackets propagating westward in the tropical troposphere ("African waves"). You may want to further explain the role of background wind, which is important for westward RG waves since they have small intrinsic group velocity. By the way, insofar as the zonal propagation of these RG wavepackets is sensitive to the background wind, it is not clear to what extent the very slight eastward displacement with altitude of their Vs variance maximum (Fig. 3b) can be interpreted simply in terms of eastward group velocity, since the winds at altitudes above 100 hPa alternate between easterly and westerly depending on the phase of the QBO. Examination of this behavior stratified by the phase of the QBO would have been helpful.

(8,4) "MRG . . . localized wave packets": Could you speculate as to why the RG waves are found only over the Atlantic and easternmost Pacific?

(8,8) "CFSR . . . has a zonally broader signal": Consistent with the spectrum shown in Fig. 2.

(8,18) "due to the data availability": I think you mean "due to the lack of ML data" for MERRA.

(8,25) "annual time series": "time series of annually-averaged data" might be clearer.

(8,32) "A similar systematic change . . . at 10 and 5 hPa": On the other hand, at 50 hPa there is no change. Any idea why? Even if you do not know, this should be pointed out.

(9,16) "the rate of change . . . is 17%": 17% is not a "rate of change"; it is the change between two periods expressed in percentage terms (note also similar, imprecise usage on line 18).

(10,11) "if duration of westerly QBO phases . . . are shorter in P2 than in P1": So, are they shorter or not? Regardless of statistical robustness, if you are going to bring this up as an explanation you should at least check—and tell the reader—whether the conjecture is true even qualitatively.

(11,14) Fz (Figs. 9 and 10. . .): The implicit assumption here is that div(F) is dominated by d(Fz)/dz. This need not be the case, especially for RG waves. In addition, you neglect stating whether the EP flux was averaged in latitude. It appears that it is, since later on (12,12) you write that Fz is averaged over $\pm 15°$. Such broad averaging can complicate the interpretation of the results; Garcia and Richter (2019) showed that averaging over a range of latitude wider than $\pm 5°$ yields misleading results for the RG waves found in their simulation of the QBO.

(11,24) "while for the Kelvin waves . . . interdependence": I do not understand what this means. Could you clarify?

(11,33) "apparently": Why apparently?

(12,12) Figure 11 . . . 15N-15S averaged": This broad latitude averaging could be problematic. See comment at (11,14).

(12,12) "Fz as function of phase speed": Note that Fz may not be the best quantity for characterizing the EP flux of RG waves. The conceptual framework assumed here appears to be that wave activity propagates from the lower to the upper stratosphere, as in a "classic" 1D model of the QBO. That is a limited perspective that might not apply

to the behavior of RG waves in the real world.

(12,30) "Kelvin wave forcing integrated…": What does the color bar at the bottom of the figure (labeled month-1) represent? How does one get, even approximately, the values quoted in this sentence from Fig. 11 plus the color bar?

(13,5) "MRG waves dissipate mainly in the lower stratosphere … zonal wind is easterly at 70 hPa": Yes, but where does the negative forcing in the descending westerly phase at 20 hPa (Fig. 11, top two rows) come from? It appears unconnected to anything below.

(13,6) "only up to 1 m s-1 month-1": This is less than a quarter of the magnitude quoted earlier for Kelvin waves. The large asymmetry in magnitude might be due to averaging over ±15°. As noted earlier, Garcia and Richter (2019) showed that averaging RG wave EP flux beyond ±5° reduces its magnitude substantially.

(13,16) "gravity waves … may play a more important role": Garcia and Richter (2019) concluded that RG EP flux divergence is much larger when averaged over a narrower range of latitude; and yet this EP flux divergence does not drive the QBO in their model but is instead a result of instability of the QBO westerlies. The implication is, indeed, that the easterly forcing must come in large part from smaller scale gravity waves.

(14,4) "suggestive of in situ wave generation": What is the generation mechanism? The idea that RG waves might be generated in situ has been proposed by Garcia and Richter (2019), who associated it with instability of the QBO westerly jet and showed that similar behavior is present in other models and in observations. However, the waves identified here do not appear to be the same as those documented by Garcia and Richter, since the latter always occur in close connection with regions where the westerly jet curvature is large, such that the barotropic vorticity gradient reverses sign. On the other hand, whatever these waves are, they might be excited by the same instability mechanism that excites the RG waves documented by Garcia and Richter. I agree that these waves merit a closer examination.

(15,2) "polarization relationships": What does this mean? Are you referring to the dispersion curves?

(15,3) "exhibit remarkably similar patterns": Perhaps you should add "in the lower stratosphere", since you showed EOF results for 50 hPa only.

(15,10) "significant changes after the late 1990s": But no changes at 50 hPa, if I am interpreting Figure 7 correctly. I have no idea why this is, but it ought to be mentioned.

---

## Author Comment (AC1) · 10 Jun 2019

$>>$ We deeply appreciate the reviewer for providing constructive comments. The manuscript is revised following the comments below. Regarding the comment #6, a new figure is added in the Supplement (Fig. S5). Please also note that the averaging latitude band for the EP flux diagnostics in Section 3.4 is changed to 5°N–5°S in response to another reviewer's comment, and one figure is added (Fig. 12). Figures A1–A3 used for the responses below are attached separately (see the bottom).

[Figure]

This paper presents analysis of stratospheric equatorial waves in several reanalysis data sets, focusing on standard space-time spectral analysis of Kelvin and Mixed Rossby Gravity waves. The analyses are mostly straightforward and the results show reasonable agreement among the reanalyses (and with previous publications), with a few outliers identified. Long-term variations in wave variances show changes likely related to input satellite data sets, in particular the transition from TOVS to ATOVS in 1998. Comparisons are also made for derived EP fluxes for the Kelvin and MRG waves. The overall results provide quantitative information on tropical wave behavior among the reanalyses, and the paper makes an original and useful contribution to the SRIP evaluations. The paper is reasonably well written and is appropriate for ACP. I have a number of mostly minor comments for the authors to consider in revision.

1) The inclusion of JRA55C (no satellite data) is especially nice for quantifying the influence of satellite data in the reanalyses, and I'm surprised at how small the differences are with JRA55 at upper levels (where radiosonde data are sparse). How should this result be interpreted, i.e. are these wave spectra mostly characteristic of the forecast model, or are upper levels constrained by lower levels?

$>>$ In the middle stratosphere ($\sim$10 hPa) where the radiosonde data are sparse, wave fields in JRA-55C are likely partly determined by the analysis of lower-level fields (20–50 hPa), when one considers that there will be upward propagation of assimilated waves, and also by the dynamics of the forecast model. Provided that the lower-level fields have been reasonably well constrained in JRA-55C (as reflected by the agreement with the other five reanalyses at $\sim$50 hPa; Figs. 1–5), the forecast-model dynamics are likely to be capable of maintaining the spectral signals of the waves at $\sim$10 hPa from below in JRA-55C as well as in JRA-55.

Characteristics of the forecast model could affect the amplitudes of waves in the middle stratosphere rather than their spectral features, given that the spectra have been

well constrained in the lower levels. The wave amplitudes might depend on the model diffusion which arises from the dynamics/numerics and vertical resolution, especially in JRA-55C with less observational constraints in the middle stratosphere. In this regard, it is seen in Fig. 8 that the difference in the wave amplitudes between JRA-55 and JRA-55C increase with height above ~10 hPa, in particular in 1999–2010 when the constraint is stronger than before in JRA-55 from the new satellite instruments.

In short, the small difference between JRA-55 and JRA-55C at ~10 hPa could be interpreted as a consequence of these constraints in the lower levels. In addition, in the early period (1981–1997), the constraint in the lower stratosphere could mostly be attributed to the radiosonde observations even in JRA-55. Additional impacts of satellite measurements seem to be minor in this period (Fig. 7), which likely result in the rather small differences in the upper levels between JRA-55 and JRA-55C (Fig. 8, left).

2) Figure 11 uses the Singapore zonal winds as a standard reference for all reanalyses. Are there any systematic differences found using the zonal winds from the separate reanalyses instead of a single reference? Does behavior of the equatorial waves in the different reanalyses 'feel' the differences in zonal winds identified in Kawatani et al (2016)?

$>>$ Yes, there are systematic differences in the altitudes of the Kelvin wave forcing, which are consistent with the differences in the zonal wind :

The zonal-mean zonal winds in each reanalysis are added in Fig. 11 in the revised manuscript. It is seen that the shear layers are located at different heights between the reanalyses, consistent with the results in Kawatani et al. (2016). The maximum Kelvin-wave forcing is found to occur at the altitudes where the wind is near zero in each reanalysis. The relevant explanation in the original manuscript [P12 L26–28] is complemented using this updated figure in the revised manuscript [P13 L16–31], and

the abstract is also updated [P1 L13–15]. The authors appreciate this comment which greatly helps to improve the paper.

3) Can you please add a right hand axis indicating period (in days) for the various spectra (Fig. 1, 2, 9,10,13). The discussions in the text are all based on wave period, and it would be helpful to simply see this on the figures.

$>>$ All the figures showing the spectra are updated following this suggestion.

4) Are there any systematic seasonal variations in the Kelvin and MRG spectra at lower levels (100 hPa) identified in the reanalyses? Since you have long records of monthly statistics it would be easy to identify such behavior.

$>>$ We calculated the symmetric and antisymmetric spectra at 100 hPa in different months and found seasonal variations in the spectra. For example, the Kelvin wave temperature variances at $k = 2$–3 seem to be largest (smallest) in boreal summer (winter) in the six reanalyses (see Fig. A1 below). The antisymmetric spectra seem to have the largest (smallest) powers in boreal winter (summer), and the low-frequency power at periods of $\sim$20 days disappear in boreal summer and early autumn (Fig. A2). However, quantitative analysis of the seasonal variations in the Kelvin and MRG waves may require a more rigorous methodology because the MJO-related and Rossby wave spectra also have very strong annual variations. When these spectral signals are strong (e.g., in boreal winter for MJO signals, Fig. A1), it becomes more complicated to define the Kelvin and MRG wave spectra properly such that their annual variations are not influenced. In this reason, we do not include the seasonal variations in the paper but focus on the climatological-mean properties and long-term variations. We hope to continue to study the seasonal variations of the equatorial waves in the future.

5) The EP flux calculations are complicated and it would be helpful to have a direct reference to the calculations used here (beyond the standard Andrews 1987 textbook). Are these calculations identical to those used in Kim and Chun (2015)? How do the climatological EP fluxes here compare to other published results?

$>>$ The method of EP flux calculation is described in more detail in the revised manuscript (Section 2) [P4 L19–31]. To identify the wave types, we used only the $(k, \omega)$-spectral filters defined in P13 L2–3 in the revised manuscript, which is probably the simplest way among the methods used in the literature (cf. Yang et al., 2003; Tindall et al., 2006; Kim and Chun, 2015). These filters are defined in the same way as those used in the previous sections for consistency. The wave separation method used in Kim and Chun (2015) is more complicated and is not applied here to facilitate comparison of the results with other studies.

The climatological vertical EP fluxes for Kelvin waves, averaged over 5°N–5°S using ERA-Interim, are 0.89 mPa at 100 hPa and 0.42 mPa at 70 hPa in our study. These values are comparable to those obtained using the method of Kim and Chun (2015): 0.87 mPa at 100 hPa and 0.37 mPa at 70 hPa. Tindall et al. (2006) applied another method based on the linear wave theory to ERA-15, and estimated the climatological zonal-momentum flux $(u'w')$ of Kelvin waves as 0.0013 and 0.0017 m$^2$ s$^{-2}$ at 100 and 70 hPa, respectively. These values can be converted approximately to vertical EP fluxes of ~0.23 and 0.20 mPa, respectively, which are 50–75% smaller than those obtained in our study. The differences in both the method and data likely cause the difference in the results.

6) The composited phase-speed spectra results are very nice (Fig. 11). For the conversion from frequency to phase-speed there is a factor of zonal wavenumber

needed to conserve spectral density (see Randel and Held, 1991, JAS). Has this been incorporated in these calculations?

$\gg$ Yes, it is incorporated such that the integral of the spectral density is conserved.

While the composite results are revealing, it would be interesting to see the variability of some of these diagnostics within the composites, for example perhaps showing a 'spaghetti plot' of the separate composited time series to see the actual variability, within and among the reanalyses. Could this be included as a Supplementary figure?

$\gg$ Thank you for this suggestion. We tried to examine the variability in the EP flux diagnostics, and could see that there are significant cycle-to-cycle variations in the EP flux diagnostics as well as in the wind profiles. We add this figure in the Supplement (Fig. S5) and include a brief statement in the revised manuscript [P14 L29–32].

7) A recent paper on observed lower stratosphere Kelvin waves is Scherrlin-Pirscher et al 2017 doi:10.5194/acp-2016-576, and this might be a useful reference to include.

$\gg$ Thank you for this information. We include this reference in the revised manuscript [P10 L24–25].

8) The high frequency disturbance in the meridional wind at upper levels has some characteristics similar to the so-called 2-day wave (zonal waves 3–4, $\sim$2 day period, occurrence in strong easterly winds); see e.g. https://doi.org/10.1029/2009JD012239. It might be useful to look at the latitude-height structure of the waves for a tentative identification.

$>>$ We agree that the spectral characteristics of the high-frequency meridional-wind disturbances at 20 hPa are similar to those of the quasi-two-day wave (QTDW), although the QTDW has appeared as a disturbance in the summer mesosphere and uppermost stratosphere in the literature. It has been proposed that the QTDW is a mixed Rossby-gravity mode triggered by the baroclinic and/or barotropic instabilities associated with the easterly jet in the summer mesosphere (e.g., Salby 1981; Pfister, 1985; McCormack et al., 2009). The 20-hPa high-frequency disturbances detected in our study may likely be related to baroclinic/barotropic instabilities as well, but associated with the easterly QBO jet, although more detailed investigations will be required to make conclusion. In any case, these disturbances may not be necessarily connected from the QTDW above, given that the occurrence of the 20-hPa disturbances is strongly tied to certain phases of the QBO, as seen in Figs. 13 and 14. We add the statements regarding the similarity in the characteristics between the 20-hPa disturbances and QTDW in the revised manuscript [P16 L12–17].

An examination of the vertical structure of the disturbances is not so simple because when the disturbances are clearly identified at 20 hPa with an easterly background wind, the spectra at the altitudes above $\sim$10 hPa (Fig. A3, left) are partly occupied by another wave signal (e.g., at 0.3 cyc/day with $k = 4$–$5$). Further analysis of the vertical structure will require more detailed investigation, which is beyond the scope of the current study.

**[ References ]**

Kim, Y.-H. and Chun, H.-Y.: Momentum forcing of the quasi-biennial oscillation by equatorial waves in recent reanalyses, Atmos. Chem. Phys., 15(12), 6577–6587,

doi:10.5194/acp-15-6577-2015, 2015.

McCormack, J. P., Coy, L., and Hoppel, K. W.: Evolution of the quasi 2-day wave during January 2006, J. Geophys. Res., 114, D20115, doi:10.1029/2009JD012239, 2009. Pfister, L.: Baroclinic instability of easterly jets with applications to the summer mesosphere, J. Atmos. Sci., 42(4), 313–330, doi:10.1175/1520-0469(1985)042<0313:BIOEJW>2.0.CO;2, 1985.

Salby, M. L.: Rossby normal modes in nonuniform background configurations. Part II: Equinox and solstice conditions, J. Atmos. Sci., 38, 1827–1840, doi:10.1175/1520-0469(1981)038<1827:RNMINB>2.0.CO;2, 1981.

Tindall, J. C., Thuburn, J., and Highwood, E. J.: Equatorial waves in the lower stratosphere. I: A novel detection method, Q. J. R. Meteorol. Soc., 132(614), 177–194, doi:10.1256/qj.04.152, 2006.

Yang, G.-Y., Hoskins, B., and Slingo, J.: Convectively coupled equatorial waves: A new methodology for identifying wave structures in observational data, J. Atmos. Sci., 60(14), 1637–1654, doi:10.1175/1520-0469(2003)060<1637:CCEWAN>2.0.CO;2, 2003.

Please also note the supplement to this comment:
https://www.atmos-chem-phys-discuss.net/acp-2019-110/acp-2019-110-AC1-supplement.pdf

[Figure]

**Supplement:**

[Figure]

**Figure A1.** The same as in Fig. 1 except for the 100-hPa spectra in different months: January–February, March–April, May–June, July–August, September–October, and November–December (JF, MA, MJ, JA, SO, and ND, respectively, from top to bottom).

[Figure]

**Figure A2.** The same as in Fig. A1 except for the spectra of the symmetric component of meridional wind. The dotted and dashed curves are the same as in Fig. 2.

[Figure]

**Figure A3.** The same as in Fig. 14 except for the spectra at 7 hPa. The composites are based on the 20-hPa zonal wind as in Fig. 14.

---

## Author Comment (AC2) · 10 Jun 2019

A very clear presentation of a thorough analysis and comparison of equatorial waves in various reanalyses. In Fig. 8, I would recommend that the same x-axis is used for the difference in T as in v.

$>>$ Thank you very much for the positive reviewer comment. The figure is updated following the recommendation.

---

## Author Comment (AC3) · 10 Jun 2019

**[ Responses to the reviewer Rolando Garcia's comments ]**

>>  We deeply appreciate the reviewer Rolando Garcia for providing suggestions and corrections. These comments greatly improved our paper.

This is a well-written, comprehensive comparison of Kelvin and Rossby-gravity waves and wave activity as represented in six reanalysis datasets. The comparison is thorough and the interpretation of the results is reasonable, as far as it goes. However, I believe there are two areas where the paper could be improved. The first (required) is the choice of latitude range over which results are averaged for Eliassen-Palm (EP) flux and EP flux divergence comparisons. Garcia and Richter (JAS, 2019) have recently shown that averaging beyond ±5° can be misleading in the case of Rossby-gravity waves because their EP flux divergence pattern changes sing within a narrow neighborhood of the Equator, such that broader latitude averaging leads to cancellation.

>>  We agree that the averaging over the broad latitude band (15°N–15°S) leads to underestimation of the EP flux divergence, especially for mixed Rossby-gravity (MRG) waves. All the EP flux diagnostics are now averaged over 5°N–5°S in the revised manuscript, following this suggestion. Please see the responses to the specific comments below.

The second (optional) would be a more thorough examination of the impact of the quasi-biennial oscillation (QBO) on the behavior of EP fluxes and wave spectra (at present, there is only one figure Fig. 11 and a short discussion thereof).

>>  More detailed analysis on the association of equatorial waves with the QBO is planned as a future study. However, we added three figures to the revised manuscript (one in this section and two in the Supplement), which complement the results presented in the original manuscript. The vertical divergence of EP flux as a function of phase speed is presented in Fig. 12, which shows that the divergence of the EP flux in the vertical direction is negligibly small for the MRG waves represented in the reanalyses. This also shows the contribution of the meridional convergence of the flux for Kelvin waves when they dissipate in the westerly shear zone of the QBO. Figure S4 shows the spectra for the meridional EP flux for antisymmetric modes, which complements the vertical EP flux spectra in Fig. 10, and Fig. S5 shows the QBO cycle-to-cycle variations in the EP flux profiles, complementing Fig. 11. In addition, by presenting the zonal-mean wind profiles for each reanalysis in Fig. 11 in response to a Reviewer #1's comment, we found that representation of the equatorial wave behaviors in the assimilated fields depends on the QBO wind in each reanalysis. This is discussed in the revised manuscript [P13 L15–31].

Otherwise, the paper is an important contribution to the literature on tropical waves, and includes a very useful discussion of the impact of new satellite observations on the reanalysis products. I believe the paper is suitable for publication once the general comments above and the specific comments listed below are addressed.

**Specific comments (page, line):**

**(4, 15)** "the EP flux formulation": The reference cited does not explain how the flux is calculated; it just gives the standard definition of EP flux. A brief description of how you go from spectral components of velocity and temperature to F(omega,k) would be helpful. Also, do you average F in latitude? Over what range? See also comment at (10,14).

>> The description for the calculation of EP flux is added in the revised manuscript [P4 L19–P5 L2].

 The EP flux diagnostics were averaged over 15°N–15°S in the original manuscript, but is now averaged over 5°N–5°S in the revised manuscript, as suggested by the review comment.

**(5,4)** "JRA-55 and JRA-55C show . . . less power below 20 hPa": Does this have anything to do with vertical resolution? Slower Kelvin waves would be prevalent in the lower stratosphere; these waves have short vertical wavelengths whose accurate representation depends on having sufficient vertical resolution. It would be useful to include in Table 1 information on the horizontal and vertical resolution of each reanalysis.

>> The 60 vertical levels of JRA-55 (and JRA-55C) are all very similar to those of ERA-Interim, with vertical spacings of 1.2–1.5 km in 100–10 hPa for both reanalyses (Fujiwara et al., 2017, Fig. 3b). Therefore, the vertical resolution seems not to be the only reason for smaller temperature amplitudes in JRA-55, although these grid spacings might be too coarse for slow Kelvin waves to be fully resolved by the model dynamics, as you point out. We include the vertical resolutions of each reanalysis in Table 1, following this suggestion. The horizontal resolutions of the six reanalyses at the equator range from 0.3° to 0.7° which we believe to be fine enough to represent the large-scale equatorial waves and are not included in the table.

**(5,6)** "thin purple" Thin purple what? Are you referring to the thin purple lines in the figure?

>> Yes. It is corrected to "thin purple contours" in the revised manuscript.

**(5,15)** "MRG generated in the region. . .": How do you know where the waves are generated?

>> There are several mechanisms for MRG wave excitation in the literature, which may be relevant for waves at different regions/altitudes. One of them is the lateral forcing by Rossby waves propagating from the extratropics (e.g., Yanai and Lu, 1983; Magana and Yanai, 1995; Kiladis et al. 2016), which occurs in the upper troposphere mostly in the western hemisphere where the westerly wind duct exists. Therefore, we consider this as a possibility of wave excitation in the upper troposphere for a portion of the MRG waves analyzed in our study, although we do not intend to specify a detailed mechanism. Indeed, the low-frequency portion of the 100-hPa spectrum shown in Fig. 2 appears mostly in the western hemisphere (Fig. S2, Section 3.2), and it is much stronger in boreal winter than in summer (not shown), indicating a strong coherence with the upper tropospheric westerly wind, although Kiladis et al. (2016) shows that such forcing can also occur outside the westerly duct.

**(5,17)** "Fig. 2, dashed": Figure 2 has many dashed lines. Do you mean the longer-dashed lines in the panels for 100 hPa?

>> Yes. In the figure caption, these lines are named dashed lines, and the other dispersion curves (for U = 0) are dotted lines. We have modified the figure caption to better reflect this.

**(5,22)** "more intense than those at lower frequencies with |k| > 4, as the altitude increases": I am not sure what this means. There are local maxima at the (omega,k) mentioned in the text at 50 and 20 hPa. At 20 hPa, these maxima are larger than any other spectral components, although this is not

the case at 50 hPa. Is that what you have in mind? I am not sure why the remark about |k|>4 is needed here.

>> Yes, that is what we meant. The sentence is now clarified [P6 L4–6].

**(6,14)** "MRG . . . wavepacket travels eastward": While this is evident from the zero-background wind dispersion relation, it may not be obvious to many readers, who are conditioned to think of RG wavepackets propagating westward in the tropical troposphere ("African waves"). You may want to further explain the role of background wind, which is important for westward RG waves since they have small intrinsic group velocity. By the way, insofar as the zonal propagation of these RG wavepackets is sensitive to the background wind, it is not clear to what extent the very slight eastward displacement with altitude of their Vs variance maximum (Fig. 3b) can be interpreted simply in terms of eastward group velocity, since the winds at altitudes above 100 hPa alternate between easterly and westerly depending on the phase of the QBO. Examination of this behavior stratified by the phase of the QBO would have been helpful.

>> Thank you for this comment. We examined the locations of MRG wave variances in different QBO phases (Fig. A1). When the background flows are moderate easterlies (U ~ –10 m s$^{-1}$) at 50 hPa (Fig. A1, left), the MRG variances at 70 and 50 hPa appear westward to the climatological-mean variances shown in Fig. 3a. The zonal displacement between 70 and 50 hPa is smaller with the moderate-easterly background flows than that in Fig. 3a, as expected. When the background flows are strong easterlies (U ~ –20 m s$^{-1}$) at 50 hPa (Fig. A1, right), the MRG variances are quite small at 50 and 70 hPa as the waves are significantly filtered out. The locations of the variances are eastward to those with moderate easterlies at 50 and 70 hPa. The displacement between 70 and 50 hPa is very small. While the locations of the MRG waves vary with the QBO phases as shown in Fig. A1, the contribution of the variances with the moderate or strong easterly flows to the climatological variances is rather small due to the filtering.

We also examined the composite averages for the 50-hPa easterly and westerly QBO phases (U < –5 m s$^{-1}$ and U > 5 m s$^{-1}$, respectively; not shown), and found a similar conclusion to the above. We add statements regarding this [P7 L23–27]. The sentence that the reviewer pointed out above is also modified [P6 L33–34].

**(8,4)** "MRG . . . localized wave packets": Could you speculate as to why the RG waves are found only over the Atlantic and easternmost Pacific?

>> While the first EOF pairs isolate the region of greatest MRG variances over the Atlantic and eastern Pacific, the higher mode EOF pairs have wave structures over the Indian Ocean and Pacific (not shown). Thus, the analysis does not imply that the MRG waves are active only over the Atlantic and easternmost Pacific. We add this information in the revised manuscript (P9 L1–2).

**(8,8)** "CFSR . . . has a zonally broader signal": Consistent with the spectrum shown in Fig. 2.

>> We agree that in the 50-hPa spectra in Fig. 2, the spectral region where the majority of the powers exist in CFSR is slightly shifted toward lower zonal wavenumbers when compared with the other reanalysis, although we did not point out this detail in the previous section. This is now stated in the manuscript [P6 L6–8; P8 L33].

**(8,18)** "due to the data availability": I think you mean "due to the lack of ML data" for MERRA.

\>\>  Thank you for the correction. It is revised as suggested [P9 L10].

**(8,25)**  "annual time series": "time series of annually-averaged data" might be clearer.

\>\>  It is revised as suggested [P9 L18].

**(8,32)**  "A similar systematic change . . . at 10 and 5 hPa": On the other hand, at 50 hPa there is no change. Any idea why? Even if you do not know, this should be pointed out.

\>\>  The 50-hPa behavior (that no change appears) is pointed out in the revised manuscript [P9 L26]. We tried to speculate about the reason in the following paragraph. We attribute the change in the 100-hPa variance to the fact that the SSU instruments do not cover this altitude whereas, after 1998, the AMSU-A instruments do. The large change in the upper stratosphere (10 and 5 hPa) is attributed to the higher vertical resolution of AMSU-A than that of SSU, where the satellite impact on the assimilated field should be larger than below due to the lack of observational constraints by radiosonde soundings. On the other hand, the 50-hPa altitude is covered by both the SSU and AMSU-A instruments as well as by the radiosonde measurement, and we suspect that this could result in the minimal effect of the satellite transition at this altitude.

**(9,16)**  "the rate of change . . . is 17%": 17% is not a "rate of change"; it is the change between two periods expressed in percentage terms (note also similar, imprecise usage on line 18).

\>\>  These are corrected in the revised manuscript [P10 L8, L10].

**(10,11)**  "if duration of westerly QBO phases . . . are shorter in P2 than in P1": So, are they shorter or not? Regardless of statistical robustness, if you are going to bring this up as an explanation you should at least check and tell the reader whether the conjecture is true even qualitatively.

\>\>  Actually, the result differs depending on how we define the zonal-wind criterion for the westerly phases (e.g., zero wind, climatological-mean wind, or some critical value for Kelvin wave propagation). Since this sentence looks rather unnecessary, we have just removed it in the revised manuscript.

**(11,14)**  Fz (Figs. 9 and 10. . .): The implicit assumption here is that div(F) is dominated by d(Fz)/dz. This need not be the case, especially for RG waves.

\>\>  We agree that the meridional convergence of the EP flux is more important than the vertical convergence for MRG waves, and that even for Kelvin waves, the meridional convergence of the flux also occurs when they dissipate. To clarify this, we added a figure that shows only the vertical convergence of the flux (Fig. 12) and included explanations in the revised manuscript [P14 L3–10; P14 L28–29]. We also added a figure showing the spectra of the meridional EP flux for antisymmetric modes (Fig. S4), while keeping the original Figs. 9 and 10 with Fz.

In addition, you neglect stating whether the EP flux was averaged in latitude. It appears that it is, since later on (12,12) you write that Fz is averaged over ±15°. Such broad averaging can complicate the interpretation of the results; Garcia and Richter (2019) showed that averaging over a range of

latitude wider than ±5° yields misleading results for the RG waves found in their simulation of the QBO.

>> In the revised manuscript, we change the averaging latitude band for the EP flux diagnostics to 5°N–5°S in Section 3.4, and the latitude band is mentioned in the first part of the section [P12 L8] as well as in Section 2 with citation of the paper [P4 L31–P5 L2].

**(11,24)** "while for the Kelvin waves . . . interdependence": I do not understand what this means. Could you clarify?

>> It is now clarified in the revised manuscript [P12 L18].

**(11,33)** "apparently": Why apparently?

>> By examing the meridional EP flux spectra during the revision process, which is now added in the Supplement (Fig. S4), we found that the reduced Fz magnitudes of the low-frequency MRG waves at 20 hPa (Fig. 10) may not come from the dissipation, given that the meridional EP flux is even strong at the low-frequency range (Fig. S4). The description of the 20-hPa antisymmetric spectra is re-written in the revised manuscript [P12 L24–28].

**(12,12)** Figure 11 . . . 15N-15S averaged": This broad latitude averaging could be problematic. See comment at (11,14).

>> In the revised manuscript, we change the averaging latitude band for the EP flux diagnostics to 5°N–5°S in Section 3.4, as also mentioned above.

**(12,12)** "Fz as function of phase speed": Note that Fz may not be the best quantity for characterizing the EP flux of RG waves. The conceptual framework assumed here appears to be that wave activity propagates from the lower to the upper stratosphere, as in a "classic" 1D model of the QBO. That is a limited perspective that might not apply to the behavior of RG waves in the real world.

>> We are aware that the Fz is not good enough to fully represent the wave activity flux for MRG waves, as the meridional EP flux often dominates over the Fz due to equatorward refraction of upward propagating waves and to in-situ generation of MRG modes. However, showing the meridional EP flux (i.e., influx into 5°N–5°S) had no added value because, by definition, it is identical to the meridional convergence of the EP flux, and the latter is found to dominate the total convergence of the EP flux for the MRG waves in the reanalyses (it is shown in Fig. 12 that the vertical EP flux divergence is very small for MRG waves everywhere). Thus, the total divergence of EP flux shown in Fig. 11 actually represents the meridional divergence, and therefore its structure is almost identical to that of the meridional EP flux itself (not shown). We add discussion regarding this in the revised manuscript [P14 L28–29]. In addition, we keep the Fz contours in Fig. 11, as we think it helps explain the upward propagating behavior of MRG waves in the lower stratosphere and it also demonstrates that the westward forcing by EP flux convergence in the westerly shear zone at 15 hPa (Fig. 11, uppermost) is not connected to that below [P14 L18–19] (see also the response to the comment (13,5) below).

**(12,30)** "Kelvin wave forcing integrated. . .": What does the color bar at the bottom of the figure (labeled month-1) represent?

>> The values presented in Fig. 11 are spectral densities of the momentum forcing with respect to the phase speed (c), of which the integral over c is the momentum forcing. Thus, their unit is per month (or per sec in the MKS system) such that the unit becomes m s$^{-1}$ month$^{-1}$ (or m s$^{-2}$) after the integral over c. For easier interpretation, the unit is now written as "m s$^{-1}$ month$^{-1}$ / (m s$^{-1}$)" in the figures in the revised manuscript.

How does one get, even approximately, the values quoted in this sentence from Fig. 11 plus the color bar?

>> At a given altitude, they can be approximated by a mean value of the spectral densities multiplied by the width of their representative phase-speed range.

**(13,5)** "MRG waves dissipate mainly in the lower stratosphere . . . zonal wind is easterly at 70 hPa": Yes, but where does the negative forcing in the descending westerly phase at 20 hPa (Fig. 11, top two rows) come from? It appears unconnected to anything below.

>> Yes, we agree that it is not connected to the waves below. They might probably be generated in situ, as the MRG waves simulated in Garcia and Richter (2019). Discussion regarding this is added to the revised manuscript [P14 L16–27].

**(13,6)** "only up to 1 m s-1 month-1": This is less than a quarter of the magnitude quoted earlier for Kelvin waves. The large asymmetry in magnitude might be due to averaging over ±15°. As noted earlier, Garcia and Richter (2019) showed that averaging RG wave EP flux beyond ±5° reduces its magnitude substantially.

>> We agree that the MRG wave forcing is underestimated when averaged beyond ±5°. All the EP flux diagnostics are now calculated in 5°N–5°S, and they result in the MRG wave forcing of ~2–3 m s$^{-1}$ month$^{-1}$ [P14 L12], but this is still much smaller than that obtained in Garcia and Richter (2019) (7.5–15 m s$^{-1}$ month$^{-1}$). We also noted the possibility of underestimation of MRG wave amplitudes in reanalyses due to the coarse vertical resolution [P15 L9–11].

**(13,16)** "gravity waves . . . may play a more important role": Garcia and Richter (2019) concluded that RG EP flux divergence is much larger when averaged over a narrower range of latitude; and yet this EP flux divergence does not drive the QBO in their model but is instead a result of instability of the QBO westerlies. The implication is, indeed, that the easterly forcing must come in large part from smaller scale gravity waves.

>> We added this information in the revised manuscript [P15 L13–14].

**(14,4)** "suggestive of in situ wave generation": What is the generation mechanism? The idea that RG waves might be generated in situ has been proposed by Garcia and Richter (2019), who associated it with instability of the QBO westerly jet and showed that similar behavior is present in other models and in observations. However, the waves identified here do not appear to be the same as those documented by Garcia and Richter, since the latter always occur in close connection with regions where the westerly jet curvature is large, such that the barotropic vorticity gradient reverses

sign. On the other hand, whatever these waves are, they might be excited by the same instability mechanism that excites the RG waves documented by Garcia and Richter. I agree that these waves merit a closer examination.

>>  The generation mechanism is not investigated in detail yet, but we agree that they could be excited by barotropic (or baroclinic) instabilities, considering the mean-wind setting for which the waves appear.

**(15,2)** "polarization relationships": What does this mean? Are you referring to the dispersion curves?

>>  It regards the phase relations between variables (i.e., coherence between the zonal wind and geopotential for Kelvin waves; 90° out-of-phase relation between the meridional wind and geopotential for MRG waves). The text is modified to reflect this [P16 L33].

**(15,3)** "exhibit remarkably similar patterns": Perhaps you should add "in the lower stratosphere", since you showed EOF results for 50 hPa only.

>>  This is added to the revised manuscript in the first part of the paragraph [P16 L27].

**(15,10)** "significant changes after the late 1990s": But no changes at 50 hPa, if I am interpreting Figure 7 correctly. I have no idea why this is, but it ought to be mentioned.

>>  It is stated in the revised manuscript that there was no change at 50 hPa [P17 L9]. Please see the response to the previous comment (8,32).

**[ References ]**

Kiladis, G. N., Dias, J., and Gehne, M.: The relationship between equatorial mixed Rossby-gravity and eastward inertio-gravity waves: Part I. J. Atmos. Sci., 73, 2123–2145, 2016.

Magaña, V. and Yanai, M.: Mixed Rossby–gravity waves triggered by lateral forcing, J. Atmos. Sci., 52, 1473–1486, doi:10.1175/1520-0469(1995)052<1473:MRWTBL>2.0.CO;2, 1995.

Yanai, M. and Lu, M.-M.: Equatorially trapped waves at the 200 mb level and their association with meridional convergence of wave energy flux, J. Atmos. Sci., 40, 2785–2803, doi:10.1175/1520-0469(1983)040<2785:ETWATM>2.0.CO;2, 1983.

**[ Figure ]**

[Figure]

**Figure A1.** The same as in Fig. 3a but for the MRG wave meridional-wind variances when (left) $U = -10 \pm 2\,\mathrm{m\,s^{-1}}$ and (right) $U = -20 \pm 2\,\mathrm{m\,s^{-1}}$, where $U$ is the 50-hPa zonal wind from radiosonde observations. The variances are normalized by the maximum value of the climatological-mean (1981–2010) variances on each horizontal plane in both panels.

---

## Author Comment (AC4) · 11 Jun 2019

The comment was uploaded in the form of a supplement:
https://www.atmos-chem-phys-discuss.net/acp-2019-110/acp-2019-110-AC4-supplement.zip